# Online characterization of primary and secondary emissions of particulate matter and acidic molecules from a modern fleet of city buses

Liyuan Zhou[1,2#], Qianyun Liu[2,a#], Christian M. Salvador[3,b], Michael Le Breton[3,c], Mattias Hallquist[3], Jian Zhen Yu[4] and Chak K. Chan[1,2*], Åsa M. Hallquist[5*]

[1] Division of Physical Sciences and Engineering, King Abdullah University of Science and Technology, Thuwal, Saudi Arabia
[2] School of Energy and Environment, City University of Hong Kong, Hong Kong SAR, China
[3] Department of Chemistry and Molecular Biology, University of Gothenburg, Gothenburg, Sweden
[4] Division of Environment and Sustainability, Hong Kong University of Science and Technology, Hong Kong, China
[5] IVL Swedish Environmental Research Institute, Gothenburg, Sweden
[a] now at: RELX Science Center, Shenzhen RELX Tech. Co,. Ltd., Shenzhen, China
[b] now at: Environmental Sciences Division, Oak Ridge National Laboratory, Oak Ridge, TN 37830, USA
[c] now at: FEV Sverige AB, Gothenburg, Sweden

[#] The authors contribute equally.

*Correspondence to*: Åsa M Hallquist (asa.hallquist@ivl.se); Chak K. Chan (chak.chan@kaust.edu.sa)

**Abstract.** The potential impact of transitioning from conventional fossil fuel to a non-fossil fuel vehicle fleet was investigated by measuring primary emissions via extractive sampling of bus plumes and assessing secondary mass formation using a Gothenburg Potential Aerosol Mass (Go:PAM) reactor from 76 in-use transit buses. Online chemical characterization of gaseous and particulate emissions from these buses was conducted using a chemical ionization mass spectrometry (CIMS) with acetate as the reagent ion, coupled with a filter inlet for gases and aerosols (FIGAERO). Acetate reagent ion chemistry selectively ionizes acidic compounds, including organic and inorganic acids, as well as nitrated and sulfated organics. A significant reduction (48-98%) in fresh particle emissions was observed in buses utilizing compressed natural gas (CNG), biodiesels like rapeseed methyl ester (RME) and hydrotreated vegetable oil (HVO), as well as hybrid-electric HVO (HVO$_{HEV}$), compared to diesel (DSL) buses. However, secondary particle formation from photooxidation of emissions was substantial across all fuel types. The median ratio of particle mass emission factors of aged to fresh emissions increased in the following order: DSL buses at 4.0, HVO buses at 6.7, HVO$_{HEV}$ buses at 10.5, RME buses at 10.8, and CNG buses at 84. Of the compounds that can be identified by CIMS, fresh gaseous emissions from all Euro V/EEV buses, regardless of fuel type, were dominated by nitrogen-containing compounds such as nitrous acid (HONO), nitric acid (HNO$_3$), and isocyanic acid (HNCO), alongside small monoacids (C$_1$-C$_3$). Notably, the emission of nitrogen-containing compounds was notably lower in Euro VI buses equipped with more advanced emission control technologies. Secondary gaseous organic acids correlated strongly with

gaseous $HNO_3$ signals ($R^2$= 0.85-0.99) in Go:PAM, but their moderate to weak correlations with post-photooxidation
secondary particle mass suggest they are not reliable tracers for secondary organic aerosol formation from bus exhaust. Our
study highlights that non-regulated compounds and secondary pollutant formation, not currently addressed in legislation, are
crucial considerations in the evaluation of environmental impacts of future fuel and engine technology shifts.
**1. Introduction**
Air pollution remains a critical global issue, posing significant threats to both human health and the environment. Despite
substantial progress in reducing emissions from major sources like industry, energy production, households, transportation,
and agriculture, the worldwide achievement of air quality targets continues to be a daunting challenge. Notably, the road
transport sector, particularly in urban environments, significantly contributes to the emissions of nitrogen oxides ($NO_x$) and
particulate matter (PM), impacting the health of individuals in densely populated regions. In tandem with these concerns,
efforts to combat climate change have spurred an increase in the adoption of renewable energy sources within the transportation
sector. Biodiesel has risen as the most prevalent renewable fuel, followed by biogas and ED95 ethanol  (Guerreiro et al., 2014).
Moreover, numerous cities are progressively integrating hybrid-electric and electric vehicles into their public transport fleets,
aiming to reduce emissions.

Emissions from vehicles, especially buses, exhibit considerable variability. They are influenced by fuel type, engine design,
operational conditions, emission after-treatment technologies and maintenance (Pirjola et al., 2016; Zhao et al., 2018; Watne
et al., 2018; Liu et al., 2019a; Zhou et al., 2020). While diesel (DSL) buses are common, there is an increasing trend towards
the use of alternative fuels such as compressed natural gas (CNG), rapeseed methyl ester (RME), and hydrotreated vegetable
oil (HVO). These alternative fuels offer several benefits, including reduced PM emissions, particularly soot, and lower levels
of carbon monoxide (CO) and total hydrocarbons (THC) (Pflaum et al., 2010; Hassaneen et al., 2012; Liu et al., 2019a).
However, the efficacy of RME and HVO in diminishing $NO_x$ emissions can be inconsistent  (Pirjola et al., 2016; Liu et al.,
2019a); and CNG buses exhibit considerable variability in particle number (PN) emissions (Watne et al., 2018). In Sweden,
approximately 23% of the fuel mix of the transport sector in 2020 comprised renewable fuels, with HVO accounting for over
half of this proportion (Vourliotakis and Platsakis, 2022; Energimyndigheten, 2021). Emission control strategies, such as
aftertreatment systems including diesel particulate filters (DPFs) and selective catalytic reduction (SCR) systems, have been
implemented to mitigate pollutant emissions from vehicles. These systems have shown significant efficacy in reducing PM
and $NO_x$ emissions respectively, though their performance can vary under different operational conditions.

Accurately determining vehicle emission factors (EFs) is crucial for devising and implementing effective air quality
policies  (Fitzmaurice and Cohen, 2022). Methods such as chassis dynamometer tests, on-board measurements with portable
emission measurement systems (PEMS), and on-road vehicle chasing experiments have been employed to assess emissions
from various types of vehicles (Kwak et al., 2014; Jezek et al., 2015; Pirjola et al., 2016). Chassis dynamometer tests offer
high repeatability over standard drive cycles but may not reflect real-world driving conditions or fleet maintenance levels.
There are also challenges in accurately replicating real-world dilution effects (Vogt et al., 2003; Kuittinen et al., 2021). On-
board measurements with PEMS provide data under a wide range of operating conditions, yet like dynamometers, they may
not realistically mimic ambient dilution processes (Giechaskiel et al., 2015; Wang et al., 2020). On-road vehicle chasing
experiments involve following individual vehicles with a mobile laboratory to capture the exhaust plumes, providing insights
into realistic dilution processes from the tailpipe to ambient air, though these experiments often require a test track to ensure
traffic safety (Wang et al., 2020; Tong et al., 2022). All three methods are limited by small sample sizes, which constrain
understanding of the real emission characteristics of vehicle fleets. Alternatively, roadside or near-road measurements provide
the ability to monitor emissions from a large number of vehicles under actual driving conditions within a short
timeframe (Hallquist et al., 2013; Watne et al., 2018; Liu et al., 2019a), which is particularly important for assessing exposure
risks to pedestrians and bus passengers. However, this method is limited by its inability to monitor specific engines or
operational conditions, such as varying engine speeds and loads. Integrating results from diverse methodologies would ideally
yield a comprehensive understanding of emissions from vehicle transport systems.

In a prior study, we conducted roadside point measurements and reported EFs for general air pollutants such as PM, $NO_x$, CO,
and THC from individual buses during stop-and-go operations at a bus stop in Gothenburg, Sweden (Liu et al., 2019a). Our
findings showed that hybrid buses, when using their combustion engines to accelerate from a standstill at bus stops, tended to
emit higher particle numbers (PN) than traditional DSL buses, likely due to their relatively smaller engines. Expanding on our
prior findings, it is important to acknowledge that primary emissions are not the only way in which engine emissions impact
air quality. Emissions from engine exhaust can contribute to secondary particles through oxidation of gas-phase species,
primarily via functionalization reactions, yielding lower-volatility products (Hallquist et al., 2009; Kroll et al., 2009).
Laboratory studies have demonstrated that secondary organic aerosols (SOA) produced from diluted vehicle exhaust frequently
exceed the levels of primary organic aerosols (POA) in less than one day of atmospheric equivalent aging (Chirico et al., 2010;
Nordin et al., 2013; Platt et al., 2013; Gordon et al., 2014b; Liu et al., 2015). Oxidation flow reactors (OFRs) enable the
simulation of several days of atmospheric aging in a few minutes, with minimized wall effects compared to traditional smog
chamber experiments (Palm et al., 2016; Bruns et al., 2015). OFRs have been extensively employed to assess the SOA
formation potential of ambient air and emissions from diverse sources, including motor exhausts (Tkacik et al., 2014; Bruns
et al., 2015; Simonen et al., 2017; Watne et al., 2018; Liu et al., 2019b; Kuittinen et al., 2021; Zhou et al., 2021; Liao et al.,
2021a; Yao et al., 2022). In real-world traffic scenarios, the rapid response capabilities and convenient deployment of OFRs,
coupled with roadside point measurements, provide a robust method for evaluating emissions from a significant number of
vehicles. This approach effectively captures the considerable variability among individual vehicles within a fleet, offering a
comprehensive view of emissions under actual driving conditions (Watne et al., 2018; Zhou et al., 2021), although it may not
encompass as extensive a range of engine operations as setups that integrate OFRs with chassis dynamometer tests (Kuittinen
et al., 2021).

Primary emissions can also be oxidized to higher-volatility products through fragmentation reactions, potentially producing
carboxylic acids (Friedman et al., 2017). Engine exhaust is a recognized primary source of organic and inorganic acids in urban
environments (Kawamura et al., 1985; Kawamura and Kaplan, 1987; Kirchstetter et al., 1996; Wentzell et al., 2013; Friedman
et al., 2017). Monocarboxylic acids are produced by both diesel and spark-ignited engines  (Zervas et al., 2001b; Crisp et al.,
2014; Zervas et al., 2001a; Kawamura et al., 1985). Recent studies have identified gaseous dicarboxylic acids in diesel
exhaust (Arnold et al., 2012), compounds likely linked to the nucleation and growth of particles (Zhang et al., 2004; Pirjola et
al., 2015). Additionally, inorganic acids such as nitric ($HNO_3$) and nitrous (HONO) acids, along with isocyanic acid (HNCO)—
implicated in serious health issues like atherosclerosis, cataracts, and rheumatoid arthritis through carbamylation reactions—
have been identified in both diesel and gasoline exhausts (Wang et al., 2007; Roberts et al., 2011; Wentzell et al., 2013; Brady
et al., 2014; Link et al., 2016; Li et al., 2021). However, the secondary production of organic acid from engine exhaust remains
poorly characterized; and it may significantly contribute to the overall organic acid budget and help explain discrepancies
between models and measurements (Paulot et al., 2011; Millet et al., 2015; Yuan et al., 2015). Furthermore, the impacts of
evolving fuel and engine technologies on emissions have not been comprehensively assessed. Recent advancements in
analytical techniques now enable simultaneous, high-resolution online measurements of both gas and particle phase acidic
species. This is facilitated by high-resolution time-of-flight chemical ionization mass spectrometry (HR-ToF-CIMS) using
acetate as the reagent ion, coupled with a filter inlet for gases and aerosols (FIGAERO) (Le Breton et al., 2019; Friedman et
al., 2017; Lopez-Hilfiker et al., 2014).

In this study, we employed the OFR Gothenburg Potential Aerosol Mass Reactor (Go:PAM) along with roadside point
measurements to capture emissions from a diverse array of fuel types and engine technologies in in-use transit buses. We
present findings on the photochemical aging of emissions from a modern fleet operating on diesel (DSL) and the latest
generation of alternative fuels, including compressed natural gas (CNG), rapeseed methyl ester (RME), and hydrotreated
vegetable oil (HVO). Our study aims to compare the secondary production of PM from individual buses in real traffic scenarios
to their primary PM emissions, examining the impact of fuel type, engine technology, and photochemical age. Furthermore,
both fresh and aged emissions of gas and particle phases are characterized using HR-ToF-CIMS, providing a comprehensive
understanding of the emissions profile and their environmental implications.

**2. Methods**
**2.1 Emission measurements**
Roadside measurements were conducted at a designated urban bus stop, featuring a bus-only lane, in Gothenburg, Sweden.
(Supporting information (SI), Figure S1). The sampling occurred from March 2nd to 12th, 2016, with the average temperature
during this period recorded at approximately 3.9°C. Extractive sampling of individual bus plumes in real traffic was used to
characterize emissions, adhering to the method outlined by Hallquist et al. (2013). Air was continuously drawn through a cord-
reinforced flexible conductive hose to the instruments housed within a nearby container. Additional details of the experimental
conditions are available in our prior publication by Liu et al. (2019a). The primary focus of this study was to utilize the OFR
Go:PAM and the HR-ToF-CIMS to explore the potential for secondary pollutant formation and to conduct a detailed chemical
characterization of both gas and particle phase compounds. An experimental schematic of the roadside sampling is shown in
Figure S2. Briefly, the emissions from passing bus plumes were characterized as they accelerated from standstill at the bus
stop. A camera was positioned at the roadside to capture bus plate numbers, facilitating bus identification and enabling the
collection of specific information on each bus, including fuel type, engine technology, and exhaust after-treatment systems.
The effective identification of emissions from individual buses was achieved by employing $CO_2$ as a tracer, as delineated by
Hak et al. (2009). The concentration of $CO_2$ was measured with a non-dispersive infrared gas analyzer (LI-840A, time
resolution 1 Hz). NO and $NO_x$ were measured with two separate chemiluminescent analyzers (Thermo Scientific™ Model 42i
$NO$-$NO_2$-$NO_x$ Analyzer). In addition, specific gaseous compounds like CO, NO, and THC, were measured using a remote
sensing device (AccuScan RSD 3000, Environmental System Products Inc.). Particle emissions were characterized using a
high time resolution engine exhaust particle sizer spectrometer (EEPS, Model 3090 TSI Inc., time resolution 10 Hz) across a
size range of 5.6-560 nm. Due to the lack of detailed knowledge about the chemical composition of the emitted particles,
particle mass calculations were based on the assumption of spherical particles of unit density.

The HR-ToF-CIMS coupled with a FIGAERO was used to derive chemical information of both gas and particle phase species.
A detailed description of the configuration of the instrument can be found elsewhere  (Aljawhary et al., 2013; Lopez-Hilfiker
et al., 2014; Le Breton et al., 2018; Le Breton et al., 2019). Acetate, employed as the reagent ion, was generated using an acetic
anhydride permeation source through a $^{210}Po$ ion source ($^{210}Po$ inline ionizer, NRD inc, Static Solutions Limited). In the ion-
molecular reaction (IMR) chamber, the gaseous sampling flow interacted with the reagent ions, leading to the ionization of
target molecules. The dual inlets of the FIGAERO enable simultaneous gas phase sampling directly into the IMR and particle
sample collection on a PTFE filter for the duration of the plume via a separate inlet. The duration of the target plume for
particle collection was indicated by particle number (PN) concentration measured by the EEPS. Once the PN concentration
reduced to undistinguishable at background levels, the filter was automatically positioned to allow the collected particles to be
evaporated into the IMR. The nitrogen flow over the filter was incrementally heated from room temperature to 200°C within
5 minutes and then maintained at this maximum temperature for 8 minutes, ensuring complete desorption of mass from the
filter, followed by analysis via HR-ToF-CIMS. Perfluoropentanoic acid (PFPA), a reliable high mass calibrant, was injected
into the CIMS inlet during the sampling period (Le Breton et al., 2019). Mass spectra were calibrated using known masses
(m/z), accurate within 4 ppm: $O_2^-$, $CNO^-$, $C_3H_5O_3^-$, $C_2F_3O_3^-$, $C_5F_9O_2^-$, $C_{10}F_{18}O_4^-$, covering a range of 32-526 m/z (more details

can be found in SI). The data were acquired at 1 s time resolution. To estimate absolute EFs, a conversion of the CIMS signal to concentration using a sensitivity factor is necessary. Based on the method of Lopez-Hilfiker et al. (2015), the maximum sensitivity was determined to be 20 Hz ppt$^{-1}$, which falls within previously reported ranges (Mohr et al., 2017). Using this maximum sensitivity provides a lower-limit estimate of EFs for all oxygenated volatile organic compounds (Zhou et al., 2021). The assumption on sensitivity did not affect the comparative analysis of EFs with respect to different fuel types.

The EFs of constituents per kilogram of fuel burnt were calculated by relating the concentration change of a specific compound in the diluted exhaust plume to the change in $CO_2$ concentration. $CO_2$ served as a tracer for exhaust gas dilution, relative to background concentration (Janhäll and Hallquist, 2005; Hak et al., 2009; Hallquist et al., 2013; Watne et al., 2018). Assumptions were made for complete combustion and carbon contents of 86.1, 77.3, 70.5, and 69.2% for DSL, RME, HVO, and CNG, respectively, were assumed (Edwards et al., 2004). Further methodological details are elaborated in Liu et al. (2019a). A more comprehensive description of the EF calculations is provided in the Supporting Information.

**2.2 Oxidation flow reactor setup**

The OFR Go:PAM was utilized for photochemical aging of emissions from individual buses to investigate the potential for secondary pollutant formation. The comprehensive description and operational protocols of the Go:PAM have been detailed previously (Watne et al., 2018; Zhou et al., 2021). Briefly, the Go:PAM is a 6.1 L continuous-flow quartz glass flow reactor with input flows such that the median residence time is approximately 37s. The reactor is equipped with two Philips TUV 30 W fluorescent lamps ($\lambda$= 254 nm) and enclosed by reflective and polished aluminium mirrors to ensure a homogeneous photon field. The UV lamps generate OH radicals through the photolysis of $O_3$ in the presence of water vapor. The relative humidity (RH) within the reactor was around 60 - 80%. The $O_3$ concentration inside the Go:PAM was measured using an ozone monitor (2B technology, model 205 dual beam ozone monitor) at around 880 ppb prior to the introduction of vehicle exhaust. Particle wall losses in the Go:PAM were corrected using size-dependent transmission efficiency (Watne et al., 2018). The OH exposure ($OH_{exp}$) inside the Go:PAM was calibrated offline using sulfur dioxide ($SO_2$), following methodologies established in previous studies (Lambe et al., 2011; Kang et al., 2007), with additional details provided in the SI. During on-road measurements, the $OH_{exp}$ may be significantly influenced by the OH reactivity (i.e., CO and HC) and titration of $O_3$ by NO in the plumes, which varied between vehicles. Thus, the OH reactivity was estimated for each bus passage using the maximum $NO_x$, CO and HC concentrations in the Go:PAM, along with corresponding water and ozone levels (Watne et al., 2018; Zhou et al., 2021). Employing the maximum concentrations of these OH- or $O_3$-consuming species represents a minimum estimate of $OH_{exp}$ in our calculations. The flow-design incorporated in the Go:PAM enables investigation of transient phenomena, such as passing plumes. It also works at relatively low ozone concentrations (less than 1 ppm), limiting reactions of other potential oxidants such as $O_3$, $NO_3$, or $O^1D$ (Zhou et al., 2021).

## 3. Results and discussion

### 3.1 Fresh and aged PM emissions from buses

The aged PM emissions ($EF_{PM:aged}$) of 133 plumes from a diverse set of buses, including 16 diesel (DSL), 11 compressed natural gas (CNG), 20 rapeseed methyl ester (RME), 20 hydrotreated vegetable oil (HVO) and 9 hybrid-electric HVO ($HVO_{HEV}$) buses, were investigated using Go:PAM. The corresponding average fresh PM emissions ($EF_{PM:Fresh}$) for these 76 buses were measured during several sequential days (Figure S2). These buses were a subset of the 234 buses described in our previous study (Liu et al., 2019a), and represent data corresponding to available Go:PAM measurements. A comprehensive discussion on the full data set for fresh condition is available in Liu et al. (2019a). Figure 1 shows the average $EF_{PM:Fresh}$ and $EF_{PM:aged}$ with respect to fuel type. Among the buses, Euro V DSL models had the highest median $EF_{PM:Fresh}$, $^{Md}EF_{PM:Fresh}$ (represented by the horizontal yellow lines), of 208 mg kg-fuel$^{-1}$, followed by $HVO_{HEV}$, RME and HVO buses with $^{Md}EF_{PM:Fresh}$ of 109, 74 and 62 mg kg-fuel$^{-1}$ respectively. CNG buses and $HVO_{HEV}$ buses equipped with a DPF under Euro VI standards exhibited the lowest $^{Md}EF_{PM:Fresh}$, with over half of these buses exhibiting $EF_{PM:Fresh}$ below the detection limit (<4.3 mg kg-fuel$^{-1}$). Except for $HVO_{HEV}$ buses with a DPF, which was limited to a small tested number, all other bus types in this subset had $^{Md}EF_{PM:Fresh}$ comparable to those of the full data set in Liu et al. (2019a), within ±30% and following the same rank order. The average EFs of fresh and aged particle emissions and general gaseous pollutants for individual buses are given in Table 1.

After photooxidation in Go:PAM, particle mass increased markedly, with half of the individual buses showing average $EF_{PM:aged}$ more than eight times their average $EF_{PM:Fresh}$. For all Euro V/ EEV buses, the median $EF_{PM:aged}$, $^{Md}EF_{PM:aged}$ (represented by the horizontal blue lines), was highest for DSL buses of 749 mg kg-fuel$^{-1}$ followed by a descending order of RME (655)> CNG (645) > HVO (543) > $HVO_{HEV}$ (509). Despite low $EF_{PM:Fresh}$, CNG buses produced substantial secondary particle mass. The DPF, proven effective in earlier studies (Martinet et al., 2017; Preble et al., 2015; May et al., 2014), efficiently reduced primary particle emissions from DSL Euro III and $HVO_{HEV}$ Euro VI buses. However, these bus types, even with DPFs, exhibited higher $EF_{PM:aged}$ than those using the same fuels but without DPFs (Euro V), albeit the number of tested buses with DPFs was limited. The variance in median $EF_{PM:aged}$ among different fuel types was less pronounced compared to $EF_{PM:Fresh}$, suggesting the presence of significant non-fuel-dependent precursor sources, such as lubrication oils and/or fuel additives (Watne et al., 2018; Le Breton et al., 2019).

Figure 2 shows the bus average $EF_{PM:Fresh}$ vs the corresponding $EF_{PM:aged}$ for individual bus passages, where the average $EF_{PM:aged}$ for each bus is indicated by a solid horizontal line. This analysis focuses on Euro V/EEV buses to ensure a sufficient number of buses in the comparison, while buses from other Euro classes were not included due to their limited numbers. The median ratio of $EF_{PM:aged}$ to $EF_{PM:Fresh}$ was highest for CNG buses (84), followed by RME (10.8), $HVO_{HEV}$ (10.5), HVO (6.7) and DSL(4.0) buses. Buses equipped with DPFs, such as DSL Euro III and $HVO_{HEV}$ Euro VI (not included in Figure 2), exhibited a median ratio exceeding 50. $EF_{PM:aged}$ exhibited notable variation between passages of the same bus, likely

attributable to emission variability between passages and different dilution levels for plumes prior to sampling into the
Go:PAM. This is illustrated in Figure 2b, where $EF_{PM:Fresh}$ and $EF_{PM:aged}$ are presented as a function of the dilution level,
indicated by the integrated $CO_2$ area. Generally, a higher integrated $CO_2$ area suggests a more concentrated plume, leading to
increased external OH and $O_3$ reactivity, which in turn reduces the concentration of OH radicals available in the Go:PAM for
precursor oxidation (Emanuelsson et al., 2013; Watne et al., 2018). Some buses displayed primary emissions too dilute for
detection (markers located to the left in Figure 2b) but still exhibited non-negligible $EF_{PM:aged}$ after oxidation. To further
examine the effects of simulated atmospheric oxidation in the Go:PAM, an estimated minimum $OH_{exp}$ was calculated for each
plume by incorporating the OH reactivities of CO and HC and the titration of $O_3$ with NO, following methodologies from
Watne et al. (2018) and Zhou et al. (2021). For all plumes, $OH_{exp}$ varied between $1.1 \times 10^9$ to $4.6 \times 10^{11}$ molecules $cm^{-3}$ s. The
$EF_{PM:aged}$ for some buses, for example, the DSL and HVO located to the right in Figure 2c, increased with increasing $OH_{exp}$.
However, due to potential large differences in the chemical composition of emissions across different passages of the same
bus, where some species are more prone to forming secondary particle mass even at lower $OH_{exp}$, the $OH_{exp}$ dependent $EF_{PM:aged}$
for other buses was less pronounced.

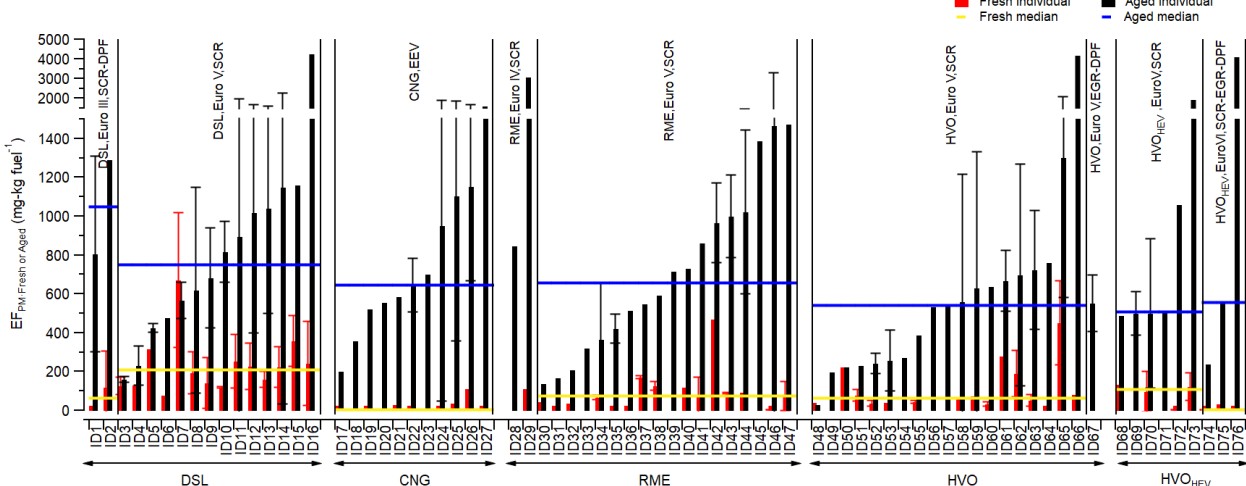


Figure 1. $EF_{PM:Fresh}$ (red bar) and $EF_{PM:aged}$ (black bar) with respect to fuel class: DSL (diesel, $ID_1$-$ID_{16}$), CNG (compressed
natural gas, $ID_{17}$-$ID_{27}$), RME (rapeseed methyl ester, $ID_{28}$-$ID_{47}$), HVO (rapeseed methyl ester, $ID_{48}$-$ID_{67}$) and $HVO_{HEV}$ (hybrid-
electric HVO, $ID_{68}$-$ID_{76}$) buses. Median values for $EF_{PM:Fresh}$ ($^{Md}EF_{PM:Fresh}$) and $EF_{PM:aged}$ ($^{Md}EF_{PM:aged}$) are indicated by
horizontal yellow and blue lines, respectively. The information on engine technology and exhaust after-treatment systems is
also shown. Given errors represent the standard deviation ($1\sigma$).

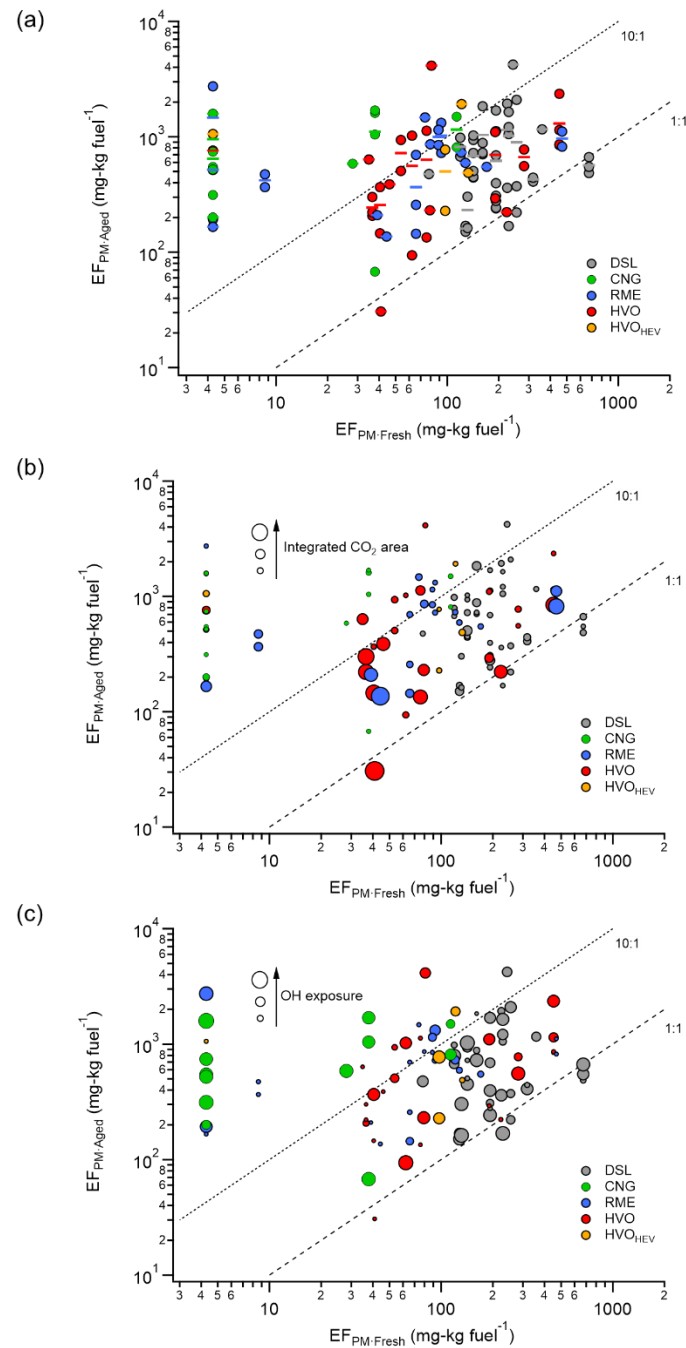



Figure 2. $EF_{PM:aged}$ vs average $EF_{PM:Fresh}$ for all the studied bus passages (Euro V) with respect to fuel type (a) and as a function
of integrated $CO_2$ area (b) and OH exposure ($OH_{exp}$) (c). The dashed lines denote the 10:1 and 1:1 $EF_{PM:aged}$: $EF_{PM:Fresh}$ ratios,
and the solid lines in (a) represent bus averages. One may note that the buses with $EF_{PM:Fresh}$ values below detection limit were
set to 4.3 mg kg-fuel[-1]. Abbreviations: DSL (diesel), CNG (compressed natural gas), RME (rapeseed methyl ester), HVO
(hydrotreated vegetable oil), HVO$_{HEV}$ (hybrid-electric HVO).

Table 1. Average particle and gaseous EFs of individual buses for fresh emissions and average EF$_{PM}$ for aged emissions[a].

| Bus ID | Fuel[c] | Euro standard | Exhaust after-treatment system[d] | EF$_{PM:Fresh}$ (mg kg$_{fuel}^{-1}$) | EF$_{PN:Fresh}$ ($10^{14}$# kg$_{fuel}^{-1}$) | EF$_{CO}$ (g kg$_{fuel}^{-1}$) | EF$_{THC}$ (g kg$_{fuel}^{-1}$) | EF$_{NOx}$ (g kg$_{fuel}^{-1}$) | EF$_{PM:Aged}$ (mg kg$_{fuel}^{-1}$) |
|---|---|---|---|---|---|---|---|---|---|
| 1 | DSL | III | SCR, DPF | 4.3 | 0.41 | 3.9±11 | 1.5±2.9 | 10±3.2 | 810±510 |
| 2 | DSL | III | SCR, DPF | 120±190 | 34±61 | 2.7±7 | 1.7±3.7 | 11±5 | 1300 |
| 3 | DSL | V | SCR | 130±45 | 3.3±1.3 | 17±18 | 0.35±1.3 | 3.9±3.7 | 160±13 |
| 4 | DSL | V | SCR | 130 | 3.6 | 20±22 | 1.5±3.6 | 4.7±7.2 | 230±100 |
| 5 | DSL | V | SCR | 320 | 5.9 | 20±28 | 2±3.5 | 9.7±7 | 430±23 |
| 6 | DSL | V | SCR | 78 | 1.6 | 20±21 | 2.7±5.6 | 13±12 | 480 |
| 7 | DSL | V | SCR | 670±350 | 10±6.8 | 42±44 | 2.3±3.7 | 6.8±5 | 570±92 |
| 8 | DSL | V | SCR | 190±110 | 6.5±3 | 14±21 | 0.75±1.7 | 12±5.1 | 620±530 |
| 9 | DSL | V | SCR | 140±130 | 4.3±2.6 | 9.8±14 | 1±1.5 | 15±13 | 680±260 |
| 10 | DSL | V | SCR | 120±4.7 | 3.2±0.66 | 16±18 | 2.5±4.7 | 12±6.9 | 820±160 |
| 11 | DSL | V | SCR | 250±140 | 4.7±2.7 | 16±23 | 0.8±1.4 | 12±8.9 | 900±1000 |
| 12 | DSL | V | SCR | 230±120 | 5.1±1.5 | 16±26 | 2.6±4.6 | 12±9.9 | 1000±620 |
| 13 | DSL | V | SCR | 160±41 | 3.5±0.97 | 27±27 | 1.4±2.7 | 17±9.8 | 1000±540 |
| 14 | DSL | V | SCR | 220±110 | 5.2±1.3 | 12±17 | 2.6±4.1 | 11±7.4 | 1100±1100 |
| 15 | DSL | V | SCR | 360±130 | 6.8±4.2 | 21±25 | 1.2±3.3 | 5.7±4.4 | 1200 |
| 16 | DSL | V | SCR | 240±220 | 22±11 | 5.5±7.5 | 0.74±1.6 | 6.8±5.6 | 4200 |
| 17 | CNG | EEV | - | 4.3±0 | 0.41±0 | n.a. | n.a. | 4.8±1.7 | 200 |
| 18 | CNG | EEV | - | n.a. | n.a. | n.a. | n.a. | 11±4.9 | 360 |
| 19 | CNG | EEV | - | 4.3±0 | 0.41±0 | n.a. | n.a. | 4±3.8 | 520 |
| 20 | CNG | EEV | - | n.a. | n.a. | n.a. | n.a. | 15±17 | 560 |
| 21 | CNG | EEV | - | 28 | 1.3 | n.a. | n.a. | 2.2±0.93 | 590 |
| 22 | CNG | EEV | - | 4.3 | 0.41 | n.a. | n.a. | 1.8±1 | 650±140 |
| 23 | CNG | EEV | - | n.a. | n.a. | n.a. | n.a. | 3.2±0.53 | 700 |
| 24 | CNG | EEV | - | 4.3 | 0.41 | n.a. | n.a. | 6.9±1.4 | 950±900 |
| 25 | CNG | EEV | - | 38 | 11 | n.a. | n.a. | 7.3±5.3 | 1100±750 |
| 26 | CNG | EEV | - | 110 | 200 | n.a. | n.a. | 8.2±4.2 | 1200±480 |
| 27 | CNG | EEV | - | 4.3±0 | 0.41±0 | n.a. | n.a. | 6±1.8 | 1600 |
| 28 | RME | IV | SCR | n.a. | n.a. | 10±8.7 | 3.1±3 | 46±20 | 850 |
| 29 | RME | IV | SCR | 110 | 4.1 | 4.2±8.4 | 0.19±0.38 | 7.2±6.8 | 3000 |
| 30 | RME | V | SCR | 44 | 2.2 | 12±14 | 2.2±3.6 | 32±32 | 140 |
| 31 | RME | V | SCR | 4.3 | 0.41 | 7.4±7.1 | 0.075±0.17 | 13±5.1 | 170 |
| 32 | RME | V | SCR | 39 | 6.2 | 5.2±4.8 | 0.87±1.1 | 18±5.4 | 210 |
| 33 | RME | V | SCR | n.a. | n.a. | 0.24±0.54 | 0.24±0.39 | 10±3.3 | 320 |
| 34 | RME | V | SCR | 66±11 | 2.4±1 | 7±7.2 | 1.8±2.7 | 23±13 | 370±290 |
| 35 | RME | V | SCR | 8.6 | 0.96 | 4.9±3.6 | 0.59±0.73 | 20±5.1 | 420±75 |
| 36 | RME | V | SCR | 4.3 | 0.41 | 22±23 | 1.8±2 | 25±16 | 520 |
| 37 | RME | V | SCR | 170±7.7 | 6.4±1 | 34±35 | 0.016±0.043 | 19±10 | 550 |
| 38 | RME | V | SCR | 130±24 | 11±14 | 17±20 | 2±4 | 16±15 | 590 |
| 39 | RME | V | SCR | n.a. | n.a. | 1.2 | 0.64 | 21 | 720 |
| 40 | RME | V | SCR | 120 | 5.3 | 12±9.4 | 1.8±2.6 | 18±8.2 | 730 |
| 41 | RME | V | SCR | 80±95 | 4.2±2.9 | 8.8±17 | 0.72±0.87 | 25±5.7 | 860 |
| 42 | RME | V | SCR | 470 | 5.8 | 4.5±5.1 | 0.23±0.38 | 18±7.8 | 970±210 |
| 43 | RME | V | SCR | 89±2.3 | 2.6±0.16 | 5.4±9.4 | 0.68±1.9 | 28±17 | 1000±210 |
| 44 | RME | V | SCR | 92 | 1.6 | 14±19 | 1.8±3 | 23±17 | 1000±420 |
| 45 | RME | V | SCR | n.a. | n.a. | 37±26 | 5.8±3.6 | 14±6.3 | 1400 |
| 46 | RME | V | SCR | 4.3±0 | 0.41±0 | 9.6±14 | 0.89±1.4 | 28±8.4 | 1500±1800 |
| 47 | RME | V | SCR | 74±75 | 12±6 | 6.1±6.3 | 1.1±1.4 | 18±5.2 | 1500 |
| 48 | HVO | V | SCR | 41 | 1.5 | 8.4±2 | 0.14±0.31 | 10±0.4 | 31 |
| 49 | HVO | V | SCR | n.a. | n.a. | 5.8±8 | 0.7±0.62 | 13±10 | 200 |
| 50 | HVO | V | SCR | 220 | 6.6 | 8.3±9.1 | 0.91±0.97 | 13±8.6 | 220 |
| 51 | HVO | V | SCR | 79±31 | 2.6±0.74 | 7.8±5.8 | 0.41±0.59 | 12±8.2 | 230 |

| 52 | HVO | V | SCR | 37±13 | 1.9±0.65 | 4.8±5.5 | 0.64±0.82 | 20±3 | 240±51 |
|---|---|---|---|---|---|---|---|---|---|
| 53 | HVO | V | SCR | 40 | 2.5 | 2.1±3.4 | 0.0083±0.019 | 16±4.3 | 260±160 |
| 54 | HVO | V | SCR | n.a. | n.a. | 2.1±3 | 0.55±0.77 | 22 | 270 |
| 55 | HVO | V | SCR | 46±6.6 | 2.6±0.52 | 6.2±4.1 | 0.79±0.55 | 12±8.2 | 390 |
| 56 | HVO | V | SCR | n.a. | n.a. | 11±10 | 0.74±0.84 | 5.7 | 530 |
| 57 | HVO | V | SCR | n.a. | n.a. | 14±17 | 0.79±1.2 | 11±2.6 | 540 |
| 58 | HVO | V | SCR | 62 | 4.1 | 6.8±6.7 | 0.22±0.31 | 11±6.3 | 560±660 |
| 59 | HVO | V | SCR | 76 | 5.3 | 2.3±2 | 0.24±0.47 | 19±3.4 | 630±700 |
| 60 | HVO | V | SCR | 35±11 | 1.5±0.19 | 3.3±5 | 0.45±0.86 | 9.2±9 | 640 |
| 61 | HVO | V | SCR | 280 | 14 | 9.9±16 | 0.55±0.73 | 11±3.6 | 670±160 |
| 62 | HVO | V | SCR | 190±120 | 68±86 | 1.1±1.9 | 0.3±0.49 | 9.3±4.9 | 700±570 |
| 63 | HVO | V | SCR | 54±30 | 4.6±2.2 | 3.5±4.6 | 0.49±0.48 | 14±3.5 | 720±310 |
| 64 | HVO | V | SCR | 4.3 | 0.41 | 2.2±3.8 | 0.33±0.73 | 12±4.8 | 760 |
| 65 | HVO | V | SCR | 450±220 | 18±18 | 1.4±1.6 | 0.28±0.37 | 12±2.6 | 1300±720 |
| 66 | HVO | V | SCR | 81 | 11 | 0.88±0.93 | 0.28±0.25 | 13±6.5 | 4100 |
| 67 | HVO | V | EGR, DPF | n.a. | n.a. | 4.6±5.9 | 0.64±1.2 | 11±8.1 | 550±150 |
| 68 | HVO$_{HEV}$ | V | SCR | 130 | 52 | 12±19 | 0.97±1.4 | 20±15 | 490 |
| 69 | HVO$_{HEV}$ | V | SCR | n.a. | n.a. | 4.1±8.4 | 0.5±1.3 | 18±3.3 | 500±110 |
| 70 | HVO$_{HEV}$ | V | SCR | 97±100 | 25±18 | 3.8±6.8 | 1.1±1.8 | 17±5.7 | 500±390 |
| 71 | HVO$_{HEV}$ | V | SCR | n.a. | n.a. | 7.6±9.9 | 2.9±2.4 | 12±2.1 | 520 |
| 72 | HVO$_{HEV}$ | V | SCR | 4.3±0 | 0.41±0 | 3.7±5.8 | 1±2.4 | 20±10 | 1100 |
| 73 | HVO$_{HEV}$ | V | SCR | 120±72 | 8.9±2.9 | 1.2±1.7 | 0.18±0.26 | 17±7 | 1900 |
| 74 | HVO$_{HEV}$ | VI | SCR,EGR,DPF | 4.3±0 | 0.41±0 | 4.7±11 | 2.2±4.7 | 7.2±8.5 | 240 |
| 75 | HVO$_{HEV}$ | VI | SCR,EGR,DPF | 33 | 29 | 1.2±2.4 | 0.22±0.49 | 6.7±3.3 | 550 |
| 76 | HVO$_{HEV}$ | VI | SCR,EGR,DPF | 4.3 | 0.41 | 10±9.2 | 1.5±2.3 | 8.8±8.7 | 4100 |

[a]Given errors represent the standard deviation (1σ).
[b]n.a., abbreviation for not available.
[c]DSL, CNG, RME, HVO and HVO$_{HEV}$, abbreviations for diesel, compressed natural gas, rapeseed methyl ester, hydrotreated vegetable oil, and hybrid-electric hydrotreated vegetable oil.
[d]SCR, DPF and EGR, abbreviations for selective catalytic reduction, diesel particulate filter and exhaust gas recirculation systems.

The secondary particle mass formed (ΔPM) was calculated as the difference between $EF_{PM:aged}$ for a plume and the average $EF_{PM:Fresh}$ for the corresponding individual bus. Figure 3 illustrates ΔPM as a function of $OH_{exp}$ for the bus fleet in this study, which includes 40% DSL, 12.2% CNG, 20% RME, 20.8% HVO, and 7% HVO$_{HEV}$. The results were grouped based on $OH_{exp}$, spanning a range from $1.1\times10^{9}$ to $4.6 \times 10^{11}$ molecules cm$^{-3}$ s. The results in this study are compared with those reported from a tunnel study (Tkacik et al., 2014), an urban roadside study of a mixed fleet in Hong Kong (Liu et al., 2019b), a depot study on rather modern types of city buses (Watne et al., 2018) and roadside measurements of a heavy-duty truck fleet in Gothenburg (Zhou et al., 2021). Laboratory OFR and chamber studies of middle-duty and heavy-duty diesel vehicles (Deng et al., 2017), diesel passenger cars (Chirico et al., 2010), a diesel engine (Jathar et al., 2017a), and gasoline vehicles (Gordon et al., 2014a; Platt et al., 2013) were also included for comparison.

The ΔPM from vehicle emissions is influenced by factors such as vehicle and fuel types, driving modes, and $OH_{exp}$ during experiments (Gentner et al., 2017). Considering the variability of OH reactivity among vehicles and the consequently wide range of $OH_{exp}$, this study, along with Watne et al. (2018), categorizes ΔPM trend into $OH_{exp}$ bins. The median ΔPM was approximately 400 mg kg-fuel$^{-1}$ at $OH_{exp} < 4.3 \times 10^{10}$ molecules cm$^{-3}$ s (corresponding to 1 OH day, assuming an OH concentration of $1 \times 10^{6}$ molecules cm$^{-3}$ for 12 h per day) and was 364-495 mg kg-fuel$^{-1}$ at 1-5 OH days, reaching a maximum

of around 920 mg kg-fuel$^{-1}$ at approximately 5-6 OH days for the bus fleet in this study. This peak value of $\Delta$PM was lower than the approximately 3000 mg kg-fuel$^{-1}$ at ~5-6 OH days observed in the depot measurements by Watne et al. (2018), a difference potentially due to variations in engine technology and fuel types used in the bus fleets. Notably, HVO was not used in the depot study, while some buses switched from RME to HVO prior to this study. The $\Delta$PM peaked and then decreased at higher $OH_{exp}$, likely due to the transition from functionalization-dominated reactions and condensation at lower $OH_{exp}$ to fragmentation reactions and evaporation dominance at higher $OH_{exp}$ (Tkacik et al., 2014; Ortega et al., 2016). The $\Delta$PM in this study was comparable to 855 mg kg-fuel$^{-1}$ for a mixed fleet consisting of 44.1% gasoline, 41.3% diesel, and 14.6% LPG vehicles measured at an urban roadside in Hong Kong (Liu et al., 2019b). It was slightly higher than the $\Delta$PM measured from a Euro VI dominated (more than 70%) heavy-duty truck fleet at an urban roadside in Gothenburg (Zhou et al., 2021), and from a fleet with over 80% light-duty gasoline vehicles in a Pittsburgh tunnel study (Tkacik et al., 2014). Additionally, the $\Delta$PM in this study was consistent with that for middle-duty and heavy-duty diesel vehicles (Deng et al., 2017), diesel passenger cars (Chirico et al., 2010), and a diesel (or biodiesel)-fuelled engine under 50% load condition (Jathar et al., 2017a) (around 190-1133 mg kg-fuel$^{-1}$). However, the diesel (or biodiesel)-fuelled engine under idle conditions can produce significantly higher $\Delta$PM (more than 5000 mg kg-fuel$^{-1}$), likely because engines at idle loads are less efficient at burning fuel, leading to higher emissions of unburnt gaseous combustion products (as precursors of secondary PM) (Nordin et al., 2013; Saliba et al., 2017; Jathar et al., 2017a). In contrast, experiments conducted for gasoline vehicles at relatively low photochemical ages (< 1 OH day) typically produced $\Delta$PM lower than 70 mg kg-fuel$^{-1}$ (Gordon et al., 2014a), except for a Euro 5 gasoline vehicle (340 mg kg-fuel$^{-1}$) operated with a New European Driving Cycle (Platt et al., 2013).

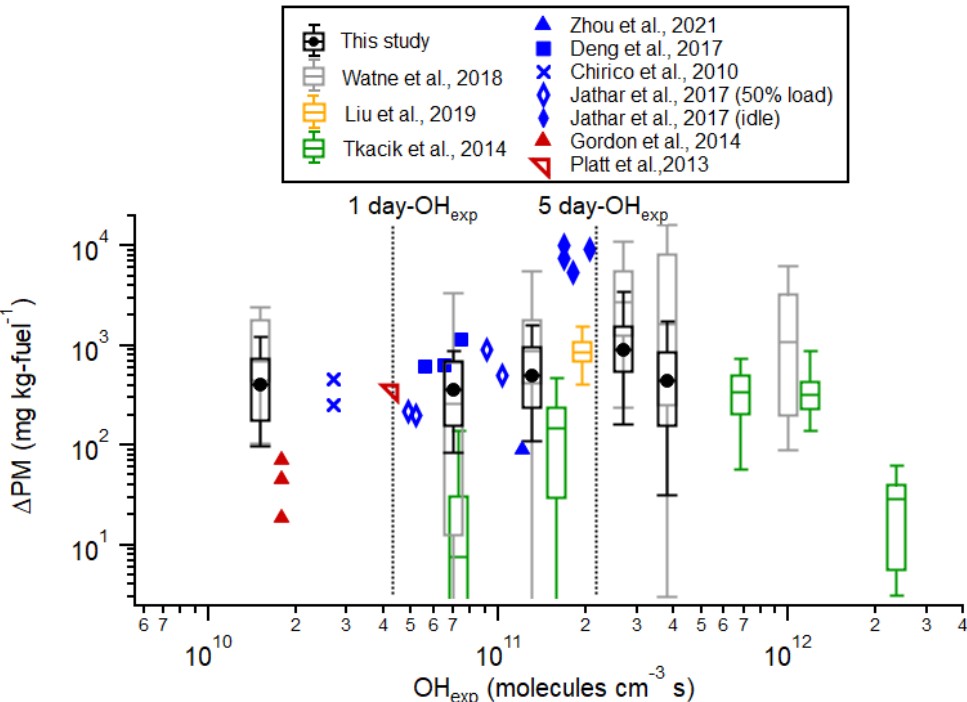



Figure 3. Secondary particle mass formed (ΔPM), calculated as $EF_{PM:aged}$ subtracted by the average $EF_{PM:Fresh}$, vs modeled OH
exposure ($OH_{exp}$) for the bus fleet in this study and comparison with those reported for a tunnel study (Tkacik et al., 2014), a
depot study (Watne et al., 2018), roadside measurements (Liu et al., 2019b; Zhou et al., 2021), middle-duty and heavy-duty
diesel vehicles (Deng et al., 2017), diesel passenger cars (Chirico et al., 2010), a diesel engine (Jathar et al., 2017a), and
gasoline vehicles (Gordon et al., 2014a; Platt et al., 2013). Dashed lines indicate 1- and 5-day $OH_{exp}$ assuming an OH
concentration of $1 \times 10^6$ molecules $cm^{-3}$ 12 h per day (Watne et al., 2018). Note that ΔPM in this study, alongside those by
Watne et al. (2018), Zhou et al. (2021) and Liu et al. (2019b), includes both secondary organic and inorganic aerosol, while
ΔPM in research by Deng et al. (2017), Chirico et al. (2010), Jathar et al. (2017a), Gordon et al. (2014a), Platt et al. (2013)
and Tkacik et al. (2014) pertains only to secondary organic aerosol mass.



**3.2 Chemical characterization using CIMS**
**3.2.1 Fresh gaseous emissions**
Figure 4 presents the median emission factors ($^{Md}EFs$) of acetate CIMS measured fresh gaseous emissions with respect to fuel
type. The identities of the organic compounds detected by HR-ToF-CIMS are assigned based on knowledge of sensitivities of
the ionization scheme and the expected compounds emitted from the buses. Plausible compounds are assigned from the

formulae, with a caveat that other isomers might contribute to the signal. These compounds were classified into nine families based on their molecular characteristics as outlined by Liu et al. (2017), with additional details provided in the SI. Among all Euro V/EEV buses, hybrid-electric HVO (HVO$_{HEV}$) buses exhibited the highest $^{Md}$EF of CIMS measured fresh gaseous emissions (68 mg kg-fuel$^{-1}$), followed by DSL (42 mg kg-fuel$^{-1}$), RME (18 mg kg-fuel$^{-1}$), and CNG (16 mg kg-fuel$^{-1}$), while HVO had the lowest $^{Md}$EF of 12 mg kg-fuel$^{-1}$. Nitrogen (N) -containing compounds (no sulfur) and monoacid families predominantly composed these fresh gaseous emissions. Compared to Euro V HVO$_{HEV}$ buses, HVO$_{HEV}$ buses equipped with exhaust gas recirculation (EGR) and DPF systems (Euro VI) demonstrated a significant reduction in $^{Md}$EF (10 mg kg-fuel$^{-1}$), primarily due to decreased emissions of N-containing compounds, although the $^{Md}$EF of other compound families were higher. In contrast, Zhou et al. (2021) reported significant reductions in both carboxylic acids and carbonyl compounds (by 94% on average), and acidic nitrogen-containing organic and inorganic species (79%) when transitioning from Euro V to Euro VI heavy-duty trucks. However, details on the types of exhaust after-treatment systems used in the trucks from such study are not specified. Moreover, this study utilized acetate as a different reagent ion for CIMS compared to the iodide used by Zhou et al. (2021). Table 2 lists the top 10 $^{Md}$EFs of fresh gaseous compounds, contributing over 88% of total fresh gaseous emissions measured by CIMS for most bus types, except for Euro VI HVO$_{HEV}$ (61%). The fresh gaseous emissions from all types of Euro V/EEV buses were primarily composed of nitrous acid (HONO) and nitric acid (HNO$_3$), with HONO being the most significant acidic emission. The $^{Md}$EFs of HONO and HNO$_3$ generally align with values reported in the literature, ranging from approximately 7-250 mg kg-fuel$^{-1}$ for HONO (Kurtenbach et al., 2001; Wentzell et al., 2013; Liao et al., 2020; Nakashima and Kondo, 2022) and around 4-14 mg kg-fuel$^{-1}$ for HNO$_3$ (Wentzell et al., 2013). Acetic acid (C$_2$H$_4$O$_2$), formic acid (CH$_2$O$_2$), and isocyanic acid (HNCO) also exhibited relatively high $^{Md}$EFs. The $^{Md}$EFs of formic acid for all Euro V/EEV bus types (0.02-1.97 mg kg-fuel$^{-1}$) were consistent with those from a light-duty gasoline fleet (0.57−0.94 mg kg-fuel$^{-1}$) reported by Crisp et al. (2014). The $^{Md}$EFs of acetic acid ranged from 1.23 to 4.84 mg kg-fuel$^{-1}$, falling between values for gasoline vehicles (0.78 mg kg-fuel$^{-1}$) and diesel buses (approximately 12-23 mg kg-fuel$^{-1}$) (Li et al., 2021). Isocyanic acid, likely an intermediate product of the thermal degradation of urea in SCR systems without sufficient hydrolysis (Bernhard et al., 2012), was detected in emissions from all bus types, with $^{Md}$EFs of 0.08-14.74 mg kg-fuel$^{-1}$. These values are slightly lower than those from a non-road diesel engine (31-56 mg kg-fuel$^{-1}$) reported by Jathar et al. (2017b), but align well with SCR-equipped diesel vehicles tested by Suarez-Bertoa and Astorga (2016) (1.3-9.7 mg kg-fuel$^{-1}$) and a diesel engine with a diesel oxidation catalyst (DOC) (Wentzell et al., 2013) (0.21-3.96 mg kg-fuel$^{-1}$). Among all Euro V/EEV buses, HVO$_{HEV}$ buses showed the highest emissions of HNCO, potentially attributed to cold engine conditions since the combustion engine does not operate continuously. Notably, emissions of HNCO were significantly lowered and neither HONO nor HNO$_3$ were identified among the top 10 $^{Md}$EFs for HVO$_{HEV}$ buses equipped with EGR and DPF systems (Euro VI), suggesting that newer engine technologies incorporating EGR and DPF systems likely effective in reducing emissions of NO$_x$ (Table 1) as well as HNCO, HONO and HNO$_3$. CH$_4$SO$_3$, potentially identified as methanesulfonic acid, was detected in the emissions from DSL and RME buses. Previous studies, such as those by Corrêa and Arbilla (2008), have shown that mercaptans, emitted from diesel and biodiesel exhausts, can transform under high NOx conditions into products including methanesulfonic acid. The presence of sulfur-

containing organic compounds in diesel fuel and lubricants, and their potential transformation upon combustion into various
sulfuric derivatives, alongside the catalytic activity of engine converters, could also contribute to such findings. However, the
detailed formation pathway of $CH_4SO_3$ in our study remains unknown.


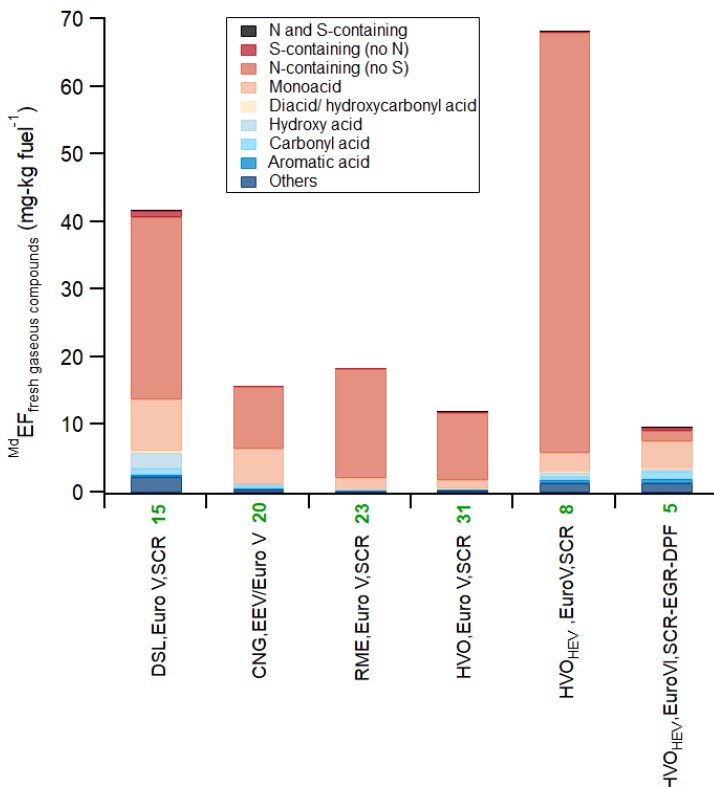


Figure 4. $^{Md}$EFs of CIMS measured fresh gaseous emissions with respect to fuel class: DSL (diesel, 15), CNG (compressed
natural gas, 20), RME (rapeseed methyl ester, 23), HVO (rapeseed methyl ester, 31) and HVO$_{HEV}$ (hybrid-electric HVO, 13)
buses. The number in bold green represents the number of buses examined.







Table 2. Summary of top 10 $^{Md}$EFs of fresh gaseous compounds measured using HR-ToF-CIMS of DSL, CNG, RME, HVO and HVO$_{HEV}$ buses[a] (color coded by different families shown in Figure 4).

| DSL, Euro V, SCR | | CNG, EEV/Euro V | | RME, Euro V, SCR | | HVO, Euro V, SCR | | HVO$_{HEV}$, Euro V, SCR | | HVO$_{HEV}$, Euro VI | |
|---|---|---|---|---|---|---|---|---|---|---|---|
| Species | $^{Md}$EF (mg kg$_{fuel}^{-1}$) | Species | $^{Md}$EF (mg kg$_{fuel}^{-1}$) | Species | $^{Md}$EF (mg kg$_{fuel}^{-1}$) | Species | $^{Md}$EF (mg kg$_{fuel}^{-1}$) | Species | $^{Md}$EF (mg kg$_{fuel}^{-1}$) | Species | $^{Md}$EF (mg kg$_{fuel}^{-1}$) |
| HONO | 20.64 | HONO | 4.92 | HONO | 12.72 | HONO | 7.62 | HONO | 38.96 | C$_3$H$_2$O$_2$ | 2.42 |
| HNO$_3$ | 5.29 | C$_2$H$_4$O$_2$ | 4.68 | HNO$_3$ | 3.24 | HNO$_3$ | 2.20 | HNCO | 14.74 | C$_2$H$_4$O$_2$ | 1.23 |
| C$_2$H$_4$O$_2$ | 4.84 | HNO$_3$ | 3.48 | C$_2$H$_4$O$_2$ | 1.23 | C$_2$H$_4$O$_2$ | 1.23 | HNO$_3$ | 7.89 | C$_2$H$_2$O$_3$ | 0.62 |
| CH$_2$O$_2$ | 1.97 | HNCO | 0.51 | CH$_2$O$_2$ | 0.48 | C$_3$H$_6$O$_3$ | 0.14 | C$_2$H$_4$O$_2$ | 1.83 | C$_8$H$_6$O$_4$ | 0.40 |
| C$_3$H$_6$O$_3$ | 1.79 | CH$_2$O$_2$ | 0.30 | HNCO | 0.15 | C$_3$H$_2$O$_2$ | 0.09 | CH$_2$O$_2$ | 0.45 | C$_6$H$_5$NO$_2$ | 0.31 |
| CH$_4$SO$_3$ | 0.71 | C$_2$H$_2$O$_3$ | 0.25 | C$_2$H$_2$O$_3$ | 0.05 | HNCO | 0.08 | C$_3$H$_6$O$_3$ | 0.43 | HNCO | 0.27 |
| HNCO | 0.67 | C$_3$H$_2$O$_2$ | 0.14 | C$_5$H$_8$O$_3$ | 0.03 | CH$_2$O$_2$ | 0.02 | C$_3$H$_2$O$_2$ | 0.34 | C$_3$H$_4$O$_5$ | 0.22 |
| C$_3$H$_4$O$_5$ | 0.37 | C$_3$H$_4$O$_2$ | 0.06 | CH$_4$SO$_3$ | 0.02 | C$_2$H$_2$O$_3$ | 0.02 | C$_9$H$_{10}$O$_3$ | 0.16 | C$_7$H$_6$O$_3$ | 0.20 |
| C$_2$H$_2$O$_3$ | 0.31 | C$_7$H$_6$O$_3$ | 0.05 | C$_3$H$_4$O$_2$ | 0.02 | C$_3$H$_4$O$_3$ | 0.02 | C$_8$H$_6$O$_4$ | 0.12 | C$_5$H$_8$O$_3$ | 0.17 |
| C$_4$H$_6$O$_4$ | 0.22 | C$_5$H$_8$O$_4$ | 0.05 | C$_6$H$_6$N$_2$O$_2$ | 0.01 | C$_4$H$_6$O$_4$ | 0.01 | C$_5$H$_8$O$_3$ | 0.10 | H$_4$N$_2$O$_2$S | 0.16 |

[a]DSL, CNG, RME, HVO and HVO$_{HEV}$, abbreviations for diesel, compressed natural gas, rapeseed methyl ester, hydrotreated vegetable oil, and hybrid-electric hydrotreated vegetable oil.

**3.2.2 Aged gaseous emissions**

Secondary carboxylic acids were measured following exposure of the exhaust to OH radicals. Figure 5 shows the correlations between ion counts of the most abundant gas-phase organic acids and nitric acid (HNO$_3$) after oxidation in the Go:PAM. HNO$_3$ serves as an indicator of NO$_x$ oxidation. Most acids exhibited both primary and secondary sources, except for dihydroxyacetic acid (C$_2$H$_4$O$_4$), which was only identified post-aging. The chemical characterization of the aged emissions was conducted on separate occasions using HR-ToF-CIMS, capturing a limited number of buses (N=19). When these buses were categorized by fuel type, the sample size for each category became smaller, constraining statistical comparison across different bus types. Nevertheless, we analyzed the relationship between various chemical species across all buses. Glycolic acid (C$_2$H$_4$O$_3$), dihydroxyacetic acid (C$_2$H$_4$O$_4$), pyruvic acid (C$_3$H$_4$O$_3$), malonic acid (C$_3$H$_4$O$_4$), lactic acid (C$_3$H$_6$O$_3$) and acetoacetic acid (C$_4$H$_6$O$_3$) showed high correlations (R$^2$= 0.85-0.99, Fig. 5a-f) with HNO$_3$ signals. In contrast, glutaric acid (C$_5$H$_8$O$_4$) and succinic acid (C$_4$H$_6$O$_4$) exhibited poorer correlations with HNO$_3$, suggesting different formation mechanisms for these two organic acids compared to the others mentioned. Notably, these two acids showed a strong correlation with each other (R$^2$= 0.97, Fig. 5i) and both belong to the diacid/hydroxycarbonyl acid families. It is important to note that many of these carboxylic acids can directly participate in secondary PM formation in the atmosphere in the presence of water vapor and a base such as ammonia (Chen et al., 2020; Huang et al., 2018; Hao et al., 2020). This process may significantly contribute to the overall secondary PM yield, reflecting a more complex interplay between gaseous emissions and particulate matter under atmospheric conditions. While most of these small organic acids correlated well with HNO$_3$, their correlations with EF$_{PM:aged}$ or ΔPM were

moderate to weak ($R^2 < 0.6$, Figure S5). This possibly indicates that the OH-driven formation of these carboxylic acids occurs on a different time scale compared to the production of organic aerosol (Friedman et al., 2017), at least in this Go:PAM experiment. This could also be due to different subsets of hydrocarbon precursors driving the production of organic acids and secondary particle mass. Similarly, Friedman et al. (2017) observed a lack of correlation between organic aerosol and gaseous organic acid concentrations downstream of the flow reactor from a diesel engine, indicating that organic acids may not be reliable tracers for secondary organic aerosol formation from diesel exhaust.

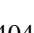

Figure 5. Correlations between ion counts of most abundant gas-phase organic acids and $HNO_3$ (a-h) and correlation between glutaric acid ($C_5H_8O_4$) and succinic acid ($C_4H_6O_4$) (i) from 19 buses after oxidation in the Go:PAM. Abbreviations: DSL (diesel), CNG (compressed natural gas), RME (rapeseed methyl ester), HVO (hydrotreated vegetable oil), HVO$_{HEV}$ (hybrid-electric HVO).

411

412

### 3.2.3 Particulate emissions

Table 3 displays the top 10 EFs of fresh particle-phase compounds ($EF_{fresh}$), as characterized by the FIGAERO ToF-CIMS, alongside their respective aged EFs ($EF_{aged}$), for Euro V DSL and RME buses. These top 10 $EF_{fresh}$ contributed to over 82% of the total fresh particulate emissions measured by CIMS. Fresh particulate emissions from DSL buses were predominantly composed of sulfuric acid ($H_2SO_4$) and nitric acid ($HNO_3$). Benzene/toluene oxidation products ($C_7H_4O_7$, $C_7H_8O$, $C_6H_5NO_3$, $C_6H_5O$, $C_7H_7NO_3$) also had relatively high $EF_{fresh}$, aligning with the findings in Le Breton et al. (2019). Similarly, high $EF_{fresh}$ of $HNO_3$ (2.5 mg kg-fuel$^{-1}$) and $H_2SO_4$ (0.61 kg-fuel$^{-1}$) were observed for the RME bus. Additionally, fatty acids, known as main components of unburned rapeseed oil (Usmanov et al., 2015), such as $C_{18}H_{34}O_2$, $C_{14}H_{28}O_2$, $C_{18}H_{36}O_2$, $C_{16}H_{32}O_2$, and $C_{16}H_{30}O_2$, significantly contributed to the identified mass loadings from the RME bus. When comparing the percentage mass observed by CIMS for both DSL and RME fuels in fresh and aged exhaust plumes, the total emission factors measured by CIMS ($EF_{CIMS}$) were notably lower than the total emission factors measured by the EEPS ($EF_{total}$). This difference is expected due to the sensitivity of the acetate ionization scheme of CIMS, which efficiently detects oxygenated volatile organic compounds, particularly carboxylic acids and inorganic acids, but has low sensitivity towards hydrocarbons and cannot detect metallic ions and soot. The CIMS measured $EF_{fresh}$ accounted for 10.4% and 5.9% of the fresh $EF_{total}$ measured by the EEPS for DSL and RME, respectively. In aged exhaust, $EF_{CIMS}$ represented a higher percentage of $EF_{total}$ (25.8% for DSL and 17.9% for RME), likely because of an increased proportion of organics with acid groups.

429

Table 3. Summary of top 10 $EF_{fresh}$ of PM contributing species with respective $EF_{aged}$ in Euro V DSL and RME emissions.

| DSL | | | RME | | |
|---|---|---|---|---|---|
| Species | $EF_{fresh}$ (mg kg$_{fuel}^{-1}$) | $EF_{aged}$ (mg kg$_{fuel}^{-1}$) | Species | $EF_{fresh}$ (mg kg$_{fuel}^{-1}$) | $EF_{aged}$ (mg kg$_{fuel}^{-1}$) |
| $H_2SO_4$ | 4.8 | 6.8 | $HNO_3$ | 2.5 | 45 |
| $HNO_3$ | 3.2 | 50 | $C_{18}H_{34}O_2$ | 1.2 | 0.81 |
| $C_7H_4O_7$ | 1.8 | 3.8 | $H_2SO_4$ | 0.61 | 0.68 |
| HNCO | 1.1 | 1.2 | $C_{14}H_{28}O_2$ | 0.52 | 0.85 |
| $C_7H_8O$ | 0.9 | 7.2 | HNCO | 0.45 | 0.089 |
| $C_3H_6O_3$ | 0.6 | 23 | $C_{18}H_{36}O_2$ | 0.32 | 0.046 |
| $C_6H_5NO_3$ | 0.53 | 2.6 | $C_{16}H_{32}O_2$ | 0.30 | 0.18 |
| $C_4H_6O_5$ | 0.45 | 0.30 | $C_6H_5O_2$ | 0.12 | 8.6 |
| $C_6H_5O$ | 0.26 | 15.6 | $C_4H_6O_4$ | 0.089 | 6.3 |
| $C_7H_7NO_3$ | 0.15 | 4.6 | $C_{16}H_{30}O_2$ | 0.081 | 0.012 |
| $EF_{total}$ measured by the EEPS | 160.9 | 1289.8 | $EF_{total}$ measured by the EEPS | 127.7 | 1320.6 |
| $EF_{CIMS}$ | 16.8 | 320.1 | $EF_{CIMS}$ | 7.5 | 237.2 |
| $EF_{CIMS}/EF_{total}$ (%) | 10.4 | 25.8 | $EF_{CIMS}/EF_{total}$ (%) | 5.9 | 17.9 |

431

432

## 4. Conclusion/ atmospheric implications

To address the challenges posed by increasing transportation needs, associated greenhouse gas emissions, and related climate change impacts, biofuels have been promoted as a low-carbon alternative to fossil fuels. In 2020, for the 27 Member States of the European Union, 93.2% of the total fuel supply for road transport was derived from fossil fuels, while 6.8% came from biofuels, with Sweden having the highest biofuel share at 23.2% (Vourliotakis and Platsakis, 2022). This study investigated renewable fuels like rapeseed methyl ester (RME), hydrotreated vegetable oil (HVO), and methane (when using biogas) in terms of primary emissions of pollutants and their secondary formation after photochemical aging. DSL buses without a DPF displayed the highest $EF_{PM:Fresh}$, whereas compressed natural gas (CNG) buses emitted the least, with a median $EF_{PM:Fresh}$ below the detection limit. Despite more than an order of magnitude difference in $EF_{PM:Fresh}$ among buses operated with various fuel types, we observed smaller variations in $EF_{PM:Aged}$, suggesting that secondary particle formation is likely influenced by substantial non-fuel-dependent precursor sources such as lubrication oils and/or fuel additives. Recognizing these sources is crucial for refining regulations on hydrocarbon emissions, which could notably enhance SOA control. The median ratios of aged to fresh particle mass emission factors, listed in ascending order, were for diesel (4.0), HVO (6.7), $HVO_{HEV}$ (10.5), RME (10.8), and CNG buses (84), highlighting the significant yet often overlooked contributions of aged/photochemically processed emissions to urban air quality. Furthermore, Zhao et al. (2017) revealed a strongly nonlinear relationship between SOA formation from vehicle exhaust and the ratio of non-methane organic gas to $NO_x$ ($NMOG:NO_x$). For instance, increasing the $NMOG:NO_x$ from 4 to 10 $ppbC/ppbNO_x$ increased the SOA yield from dilute gasoline vehicle exhaust by a factor of 8, underscoring the importance of integrated emission control policies for $NO_x$ and organic gases for better manage SOA formation. While implementing regulations for secondary particle formation presents significant challenges, these are crucial for a thorough understanding of their impact on regional air quality and health. Our approach to measuring the maximum SOA formation potential—peaking at a photochemical age of approximately 5 equivalent days of atmospheric OH exposure—provides a possible semi-quantitative reference for comparing SOA formation potential across different studies. We acknowledge the limitations of this approach for direct regulatory application and emphasize the need for more precise and comprehensive research to develop a methodologically robust framework that stakeholders can agree upon for systematically assessing the impacts of vehicle on air quality and informing regulatory strategies.

It is important to note that the ambient temperature during this study was relatively low, which does not affect the EF comparison across different buses but should be aware of when comparing these results to studies conducted at significantly higher temperatures. Wang et al. (2017) noted lower particle number EFs in summer compared to winter, potentially due to increased nucleation or condensation at cooler temperatures. Temperature impacts on emissions are significant during cold starts when combustion is inefficient (Nam et al., 2010). Post-warm-up, soot mode particles show little temperature sensitivity (Ristimäki et al., 2005). Book et al. (2015) found inconsistent trends in particle emissions from DPF-equipped

diesel trucks across various temperatures and driving cycles, suggesting that more research is needed to understand the
temperature effects on emissions from different bus types under varied operational conditions.

Non-regulated chemical species can also have serious negative impacts on air quality and human health. Organic and inorganic
acids influence the pH of precipitation and will potentially contribute to acid deposition, affecting ecosystem health.
Furthermore, there is a risk that some abatement systems might generate unintended compounds, such as HNCO from the
thermal degradation of urea in SCR systems without sufficient hydrolysis. Additionally, Jathar et al. (2017b) observed
substantial direct emissions of HNCO from diesel engines and estimated that ambient concentrations in Los Angeles could
vary widely, ranging from 20 to 107 ppt depending on different parameterizations of diesel engine emissions. The persistence
of HNCO in the atmosphere, particularly under dry conditions, poses significant health risks. It has been linked to severe
outcomes including respiratory and cardiovascular disorders, atherosclerosis, cataracts, and rheumatoid arthritis (Leslie et al.,
2019; Roberts et al., 2011). In our study, small monoacids ($C_1$-$C_3$) and nitrogen-containing compounds, such as nitrous acid
(HONO), nitric acid ($HNO_3$), and HNCO, dominated the fresh gaseous emissions measured by acetate-CIMS for all Euro
V/EEV buses regardless of fuel type, with $HVO_{HEV}$ buses exhibiting the highest emissions. Notably, the emission levels of
nitrogen-containing compounds were significantly lowered in Euro VI buses, equipped with advanced after-treatment systems
that include EGR and DPF technologies in addition to SCR-only techniques. This indicates that transitioning to vehicles
equipped with more advanced emission control technologies can be beneficial, even though these technologies may not be
specifically designed to target emissions of HONO, $HNO_3$, and HNCO. Consequently, a detailed evaluation of the
environmental and health effects of emerging engine and after-treatment technologies is highly desirable for future
considerations. Overall, the extended online chemical characterization of in-use fleet emissions, utilizing advanced techniques
like HR-ToF-CIMS, enables the identification of unregulated pollutants, which is crucial for more informed policy decisions
and vehicle technology developments.

*Data availability.*
The data used in this publication are available to the community and can be accessed by request to the corresponding author.

*Author contributions.*
ÅMH, MLB and QL conducted the measurements. ÅMH designed the project, coordinated the measurements and together
with MH and CKC supervised the study. LZ, QL, MLB, CMS and ÅMH carried out the data analysis. LZ, QL, JZY, MH,
ÅMH and CKC prepared the manuscript. All co-authors contributed to the discussion and the interpretation of the results.

*Competing interests.*
The authors declare that they have no known competing financial interests or personal relationships that could have appeared
to influence the work reported in this paper.

*Acknowledgments.*
This work was financed by VINNOVA, Sweden's Innovation Agency (2013-03058) and Formas (2020-1907) and was an
initiative within the framework programme "Photochemical smog in China" financed by the Swedish Research Council (639-
2013-6917). Chak K. Chan would like to acknowledge the support of the National Natural Science Foundation of China
(project no. 41675117 and 41875142).

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
