# Peer review of "Online characterization of primary and secondary emissions of"

_EGUsphere, 2024_

## Author Comment (AC1)

**Responses to reviewers**

Reviewer comments are in in ***black italic type***. Author responses are labelled with [R] and authors actions with [A]. Line numbers in the responses correspond to the **revised manuscript with track-changes**. Modifications to the manuscript are in blue.

**Reviewer 1**

*The paper reports on measurements of particulate and gaseous compounds in primary emissions and secondary mass formation in the exhaust plumes of transit buses that utilize different fuels and after-treatment systems. The primary focus of this study to explore the potential for secondary pollutant formation using an oxidative flow reactor and to conduct a detailed chemical characterization of both gas and particle phase compounds.*

*The study is interesting and presents comprehensive data on primary emissions of buses and composition of aged exhaust. With regards to the novelty of methods the study should be published in Atmospheric Chemistry and Physics. However, the employed methodologies for exhaust sampling and determining particle mass emission factors of fresh and aged emissions are difficult to evaluate based on the provided information. The paper lacks an explanation of the regulatory purpose of secondary particle mass (Delta-PM) and how such a regulation can be implemented across engines and fuels given the wide variation of oxidation conditions in the oxidative flow reactor.*

[R0] We appreciate the reviewer's valuable feedback and constructive suggestions.

[A0] To address concerns regarding the methodologies used in our study, we have improved the manuscript with additional details on exhaust sampling and the calculation of emission factors for various pollutants for each bus. After line 175, we have added: 'Roadside measurements were conducted at a designated urban bus stop, featuring a bus-only lane, in Gothenburg, Sweden. (Supporting information (SI), Figure S1). Extractive sampling of individual bus plumes in real traffic was used to characterize emissions, adhering to the method outlined by Hallquist et al. (2013). Air was continuously drawn through a cord-reinforced flexible conductive hose to the instruments housed within a nearby container. Additional details of the experimental conditions are available in our prior publication by Liu et al. (2019a).'

And after line 219: 'The EFs of constituents per kilogram of fuel burnt were calculated by relating the concentration change of a specific compound in the diluted exhaust plume to the change in $CO_2$ concentration. $CO_2$ served as a tracer for exhaust gas dilution, relative to background

concentration (Hallquist et al., 2013; Janhäll and Hallquist, 2005; Watne et al., 2018; Hak et al., 2009). Assumptions were made for complete combustion and carbon contents of 86.1, 77.3, 70.5, and 69.2% for DSL, RME, HVO, and CNG, respectively, were assumed (Edwards et al., 2004). Further methodological details are elaborated in Liu et al. (2019a). A more comprehensive description of the EF calculations is provided in the Supporting Information.'

We have added in the Supporting Information the detailed methodology for calculating the emission factors (EFs) of gaseous and particulate emissions for individual buses:

'Calculation of emission factors (EFs)

The EFs are calculated as the quantity of pollutant emitted per kilogram of fuel combusted, employing the carbon balance method adapted from established methodologies in our prior work and other studies in the field (Ban-Weiss et al., 2009; Hak et al., 2009; Hallquist et al., 2013; Zhou et al., 2020):

$$EF_{pollutant} = \left(\frac{\int_{t_1}^{t_2}([pollutant]_t - [pollutant]_{t_1})dt}{\int_{t_1}^{t_2}([CO_2]_t - [CO_2]_{t_1})dt}\right) \times EF_{CO_2}, \qquad (1)$$

where $EF_{pollutant}$ denotes the emission factor for the specific pollutant. The time interval from $t_1$ to $t_2$ is critical for capturing the emission dynamics from individual buses. This interval starts just before the sharp rise in pollutant concentration ($t_1$) and ends as levels stabilize and blend with ambient conditions ($t_2$), as illustrated in Figure S4, similar to methods we have described previously (Zhou et al., 2020; Zhou et al., 2021). This approach ensures the robustness of peak integration, given that contributions beyond $t_2$ typically oscillate around zero. The start and end times, $t_1$ and $t_2$, for each pollutant peak are independently determined to account for the differing response times of the measuring instruments to the exhaust plume. The resultant concentration changes of the pollutant and $CO_2$, recorded between $t_1$ and $t_2$, are integral to this calculation. For $EF_{CO_2}$, specific values are assigned based on the type of fuel used, consistent with values reported in earlier studies: 3156 g kg$^{-1}$ for DSL, 2834 g kg$^{-1}$ for RME, 3107 g kg$^{-1}$ for HVO and 2536 g kg$^{-1}$ for CNG (Edwards et al., 2004).'

[Figure]

**Figure S4.** Representative concentration profiles of $CO_2$, NOx, particle number (PN), particle mass (PM), and particle number size distribution measured from three individual buses.

A regulatory framework for secondary particle mass ($\Delta$PM) would serve a crucial role in addressing the significant yet often overlooked contributions of aged/photochemically processed emissions to PM in urban air quality. $\Delta$PM captures the additional PM formed after emissions have undergone atmospheric aging processes, reflecting their potential environmental impact more accurately than fresh emissions alone.

Implementing $\Delta$PM regulations across various engine types and fuel compositions is indeed challenging, given that the contribution of on-road vehicles to atmospheric SOA is influenced by multiple factors, including precursor concentrations and atmospheric chemistry (such as SOA yield and $OH_{exp}$). Our study observed smaller variations in $EF_{PM:Aged}$ compared to $EF_{PM:Fresh}$ across different fuel types, likely due to significant non-fuel-dependent precursor sources such as lubrication oils and/or fuel additives. This highlights the necessity of thoroughly understanding all potential precursor emissions—including those from fuels, lubrication oils, and fuel additives—to enhance current regulations on hydrocarbon emissions. Such improvements could substantially benefit the control of SOA formation. Furthermore, the study by Zhao et al. (2017) demonstrated a strongly nonlinear relationship between SOA formation from vehicle exhaust and the ratio of non-methane organic gas to $NO_x$ (NMOG:$NO_x$), significantly affecting peroxy radicals. For example, increasing the NMOG:$NO_x$ from 4 to 10ppbC/ppbNO$_x$ increased the SOA yield from dilute gasoline vehicle exhaust by a factor of 8, underscoring the importance of integrated emission control

policies for NOx and organic gases for better control SOA formation. These findings advocate for a detailed understanding and regulatory consideration to manage SOA formation more effectively. Our approach to measuring the maximum SOA formation potential—peaking at a photochemical age of approximately 5 equivalent days of atmospheric OH exposure—provides a possible semi-quantitative reference for comparing SOA formation potential across different studies. We acknowledge the limitations of this approach for direct regulatory application and emphasize the need for more precise and comprehensive research to develop a methodologically robust framework that stakeholders can agree upon for systematically assessing the impacts of vehicle on air quality and informing regulatory strategies.

In response to these findings, we have included the following discussion in our manuscript after line 512: 'Despite more than an order of magnitude difference in $EF_{PM:Fresh}$ among buses operated with various fuel types, we observed smaller variations in $EF_{PM:Aged}$, suggesting that secondary particle formation is likely influenced by substantial non-fuel-dependent precursor sources such as lubrication oils and/or fuel additives. Recognizing these sources is crucial for refining regulations on hydrocarbon emissions, which could notably enhance SOA control. The median ratios of aged to fresh particle mass emission factors, listed in ascending order, were for diesel (4.0), HVO (6.7), $HVO_{HEV}$ (10.5), RME (10.8), and CNG buses (84), highlighting the significant yet often overlooked contributions of aged/photochemically processed emissions to urban air quality. Furthermore, Zhao et al. (2017) revealed a strongly nonlinear relationship between SOA formation from vehicle exhaust and the ratio of non-methane organic gas to $NO_x$ ($NMOG:NO_x$). For instance, increasing the $NMOG:NO_x$ from 4 to 10 ppbC/ppbNO$_x$ increased the SOA yield from dilute gasoline vehicle exhaust by a factor of 8, underscoring the importance of integrated emission control policies for $NO_x$ and organic gases for better manage SOA formation. While implementing regulations for secondary particle formation presents significant challenges, these are crucial for a thorough understanding of their impact on regional air quality and health. Our approach to measuring the maximum SOA formation potential—peaking at a photochemical age of approximately 5 equivalent days of atmospheric OH exposure—provides a possible semi-quantitative reference for comparing SOA formation potential across different studies. We acknowledge the limitations of this approach for direct regulatory application and emphasize the need for more precise and comprehensive research to develop a methodologically robust framework that stakeholders can agree upon for systematically assessing the impacts of vehicle on air quality and informing regulatory strategies.'

*1.) Introduction (P3, line 77-82): Mention that dicarboxylic acids have been found in the primary emissions of diesel vehicles (Arnold et al., 2012) and that they are possible candidates for nucleation of particles in diesel exhaust, please refer to Pirjola et al. (2015).*

[R1] Thanks for the suggestion.

[A1] We have revised the introduction to include pertinent references that highlight the relevance of dicarboxylic acids in diesel emissions. The revised text now reads: 'Primary emissions can be oxidized to higher-volatility products through fragmentation reactions, potentially producing carboxylic acids (Friedman et al., 2017). Engine exhaust is a recognized primary source of organic and inorganic acids in urban environments (Kawamura et al., 1985; Kawamura and Kaplan, 1987; Kirchstetter et al., 1996; Wentzell et al., 2013; Friedman et al., 2017). Monocarboxylic acids are produced by both diesel and spark-ignited engines (Zervas et al., 2001b; Crisp et al., 2014; Zervas et al., 2001a; Kawamura et al., 1985). Recent studies have identified gaseous dicarboxylic acids in diesel exhaust (Arnold et al., 2012), compounds likely linked to the nucleation and growth of particles (Pirjola et al., 2015; Zhang et al., 2004).'

*2.) Introduction (P3, line 85-92): Suggest a separate paragraph on oxidation flow reactors. Their use in connection with point sampling at street locations should be explained in more detail. For example, the cited study by Kuittinen et al. (2021) uses the OFR to assess secondary aerosol production during standard driving cycles and real-world driving cycles. The combination with point sampling seems to be relatively new and the in-plume measurement at a single point is not well characterized compared to the measurement in driving cycles where the diluted exhaust is directly introduced into an oxidation flow reactor.*

[R2] Thank you for the valuable suggestion.

[A2] We have expanded our manuscript to clarify the distinctions among various emission measurement methods, including roadside point sampling, chassis dynamometer measurements with different driving cycles, on-road vehicle chasing, and on-board measurements with portable emission measurement systems (PEMS). After line 76, we have added: 'Accurately determining vehicle emission factors (EFs) is crucial for devising and implementing effective air quality policies (Fitzmaurice and Cohen, 2022). Methods such as chassis dynamometer tests, on-board measurements with portable emission measurement systems (PEMS), and on-road vehicle chasing experiments have been employed to assess emissions from various types of vehicles (Kwak et al., 2014; Jezek et al., 2015; Pirjola et al., 2016). Chassis dynamometer tests offer high repeatability over standard drive cycles but may not reflect real-world driving conditions or fleet maintenance levels. There are also challenges in accurately replicating real-world dilution effects (Vogt et al., 2003; Kuittinen et al., 2021). On-board measurements with PEMS provide data under a wide range of operating conditions, yet like dynamometers, they may not realistically mimic ambient dilution processes (Wang et al., 2020; Giechaskiel et al., 2015). On-road vehicle chasing experiments involve following individual vehicles with a mobile laboratory to capture the exhaust plumes, providing insights

into realistic dilution processes from the tailpipe to ambient air, though these experiments often require a test track to ensure traffic safety (Wang et al., 2020; Tong et al., 2022). All three methods are limited by small sample sizes, which constrain understanding of the real emission characteristics of vehicle fleets. Alternatively, roadside or near-road measurements provide the ability to monitor emissions from a large number of vehicles under actual driving conditions within a short timeframe (Hallquist et al., 2013; Liu et al., 2019a; Watne et al., 2018), which is particularly important for assessing exposure risks to pedestrians and bus passengers. However, this method is limited by its inability to monitor specific engines or operational conditions, such as varying engine speeds and loads. Integrating results from diverse methodologies would ideally yield a comprehensive understanding of emissions from vehicle transport systems. In a prior study, we conducted roadside point measurements and reported EFs for general air pollutants such as PM, $NO_x$, CO, and THC from individual buses during stop-and-go operations at a bus stop in Gothenburg, Sweden (Liu et al., 2019a). Our findings showed that hybrid buses, when using their combustion engines to accelerate from a standstill at bus stops, tended to emit higher particle numbers (PN) than traditional DSL buses, likely due to their relatively smaller engines.'

Furthermore, to directly address your point about the application of OFRs, we have included a separate paragraph in the introduction detailing their use in combination with point sampling at street locations after Line 120: 'Expanding on our prior findings, it is important to acknowledge that primary emissions are not the only way in which engine emissions impact air quality. Emissions from engine exhaust can contribute to secondary particles through oxidation of gas-phase species, primarily via functionalization reactions, yielding lower-volatility products (Hallquist et al., 2009; Kroll et al., 2009). Laboratory studies have demonstrated that secondary organic aerosols (SOA) produced from diluted vehicle exhaust frequently exceed the levels of primary organic aerosols (POA) in less than one day of atmospheric equivalent aging (Nordin et al., 2013; Platt et al., 2013; Gordon et al., 2014; Liu et al., 2015; Chirico et al., 2010). Oxidation flow reactors (OFRs) enable the simulation of several days of atmospheric aging in a few minutes, with minimized wall effects compared to traditional smog chamber experiments (Palm et al., 2016; Bruns et al., 2015). OFRs have been extensively employed to assess the SOA formation potential of ambient air and emissions from diverse sources, including motor exhausts (Simonen et al., 2017; Kuittinen et al., 2021; Bruns et al., 2015; Tkacik et al., 2014; Watne et al., 2018; Liu et al., 2019b; Yao et al., 2022; Zhou et al., 2021). In real-world traffic scenarios, the rapid response capabilities and convenient deployment of OFRs, coupled with roadside point measurements, provide a robust method for evaluating emissions from a significant number of vehicles. This approach effectively captures the considerable variability among individual vehicles within a fleet, offering a comprehensive view of emissions under actual driving conditions (Watne et al., 2018; Zhou et al., 2021; Liao et al., 2021; Ghadimi et al., 2023), although it may

not encompass as extensive a range of engine operations as setups that integrate OFRs with chassis dynamometer tests (Kuittinen et al., 2021).'

We have also revised the description after Line 164 as follows: 'In this study, we employed the OFR Gothenburg Potential Aerosol Mass Reactor (Go:PAM) along with roadside point measurements to capture emissions from a diverse array of fuel types and engine technologies in in-use transit buses. We present findings on the photochemical aging of emissions from a modern fleet operating on diesel (DSL) and the latest generation of alternative fuels, including compressed natural gas (CNG), rapeseed methyl ester (RME), and hydrotreated vegetable oil (HVO). Our study aims to compare the secondary production of PM from individual buses in real traffic scenarios to their primary PM emissions, examining the impact of fuel type, engine technology, and photochemical age. Furthermore, both fresh and aged emissions of gas and particle phases are characterized using HR-ToF-CIMS, providing a comprehensive understanding of the emissions profile and their environmental implications.'

**3.) Methods (P4, line 104-106) It remains uncertain how well the sampling represents concentrations in the bus plume. Does the sampling integrate over the horizontal and vertical expansion of the plume? Please discuss how temperature, wind speed and direction affected the emission factor of fresh PM.**

[R3] We apologize for any confusion caused.

[A3] To calculate the emission factors, we determined particle and gaseous emissions from individual vehicles by measuring changes in concentration, reflecting the integrated concentration over the entire pollutant peak, in the diluted exhaust plume relative to pre-passage levels and changes in $CO_2$ concentration. For more details, please refer to [R0]. This method does not require absolute concentration measurements as the ratio to $CO_2$ is assumed constant during dilution (Hak et al., 2009; Hallquist et al., 2013), effectively compensating for varying degrees of dilution. Thus, factors such as wind speed and direction do not affect the emission factors of fresh PM.

Our campaign was conducted in the winter from March 2nd to 12th, 2016, in Gothenburg, Sweden, where average daytime temperatures ranged from 2 to 5°C, with an overall average of 3.9°C. These conditions are unlikely to affect the comparative analysis of EFs for different buses within this study. However, caution is advised when directly comparing these results to data obtained under significantly higher ambient temperatures. Wang et al. (2017) observed that fleet average particle number EFs were lower in summer than in winter, likely due to enhanced nucleation or condensation of gaseous organics at lower temperatures, as opposed to summer conditions that tend to favor particle evaporation. The impact of temperature on

vehicle emissions is most pronounced during the initial cold start phase, where vehicles and their catalytic converters, still cold, lead to inefficient combustion and potentially operate under fuel-rich conditions (Nam et al., 2010). On the other hand, Ristimäki et al. (2005) found that soot mode particles are relatively unaffected by ambient temperatures post-engine warm-up. Additionally, Book et al. (2015) examined temperature effects on particulate emissions from DPF-equipped diesel trucks operating on conventional diesel and biodiesel fuels, finding that particle number and mass emission factors showed varying trends depending on the driving cycles, and concluded that temperature effects were inconclusive. In light of this, the following has been added to the manuscript after line 175: 'Roadside measurements were conducted at a designated urban bus stop, featuring a bus-only lane, in Gothenburg, Sweden. (Supporting information (SI), Figure S1). The sampling occurred from March 2nd to 12th, 2016, with the average temperature during this period recorded at approximately 3.9°C.' and after line 531, 'It is important to note that the ambient temperature during this study was relatively low, which does not affect the EF comparison across different buses but should be aware of when comparing these results to studies conducted at significantly higher temperatures. Wang et al. (2017) noted lower particle number EFs in summer compared to winter, potentially due to increased nucleation or condensation at cooler temperatures. Temperature impacts on emissions are significant during cold starts when combustion is inefficient (Nam et al., 2010). Post-warm-up, soot mode particles show little temperature sensitivity (Ristimäki et al., 2005). Book et al. (2015) found inconsistent trends in particle emissions from DPF-equipped diesel trucks across various temperatures and driving cycles, suggesting that more research is needed to understand the temperature effects on emissions from different bus types under varied operational conditions.'

**4.) The aged PM from different measurements seems to be hardly comparable because of a wide range of different OH exposure in the chamber. Were there any controls made for ageing PM to ensure that no desorption from the wall or losses on the walls affects it? Lubrication oils might substantially contribute to aged PM, which do not depend on the type of fuel.**

[R4] [A4] In our study, the transmission efficiency for particles with mobility diameters (dm) larger than 25 nm was over 90% in the Go: PAM, as characterized using nebulized ammonium sulfate particles (Watne et al., 2018). We accounted for potential losses by correcting with size-dependent transmission efficiency, as detailed at line 237: 'particle wall losses in the Go:PAM were corrected using size-dependent transmission efficiency (Watne et al., 2018).'

We agree that lubrication oils could significantly contribute to aged PM, independently of the fuel type used. Consequently, we have incorporated additional discussions into our manuscript to address this, indicated after line 273: 'The variance in median $EF_{PM:aged}$ among different fuel types was less pronounced

compared to $EF_{PM:Fresh}$, suggesting the presence of significant non-fuel-dependent precursor sources, such as lubrication oils and/or fuel additives (Watne et al., 2018; Le Breton et al., 2019).'

**5.) Page 7, line 209-210: The explanation for the less pronounced dependence of EF_PM:aged on OHexp for other buses is unclear. Potential large differences in emissions and dilution effects are already reflected in the fresh PM, how can that affect the dependence of aged PM on OHexp?**

[R5] We apologize for any confusion caused.

[A5] To clarify the points raised, we have revised the paragraph in line 292 as follows: 'Figure 2 shows the bus average $EF_{PM:Fresh}$ vs the corresponding $EF_{PM:aged}$ for individual bus passages, where the average $EF_{PM:aged}$ for each bus is indicated by a solid horizontal line. This analysis focuses on Euro V/EEV buses to ensure a sufficient number of buses in the comparison, while buses from other Euro classes were not included due to their limited numbers. The median ratio of $EF_{PM:aged}$ to $EF_{PM:Fresh}$ was highest for CNG buses (84), followed by RME (10.8), $HVO_{HEV}$ (10.5), HVO (6.7) and DSL(4.0) buses. Buses equipped with DPFs, such as DSL Euro III and $HVO_{HEV}$ Euro VI (not included in Figure 2), exhibited a median ratio exceeding 50. $EF_{PM:aged}$ exhibited notable variation between passages of the same bus, likely attributable to emission variability between passages and different dilution levels for plumes prior to sampling into the Go:PAM. This is illustrated in Figure 2b, where $EF_{PM:Fresh}$ and $EF_{PM:aged}$ are presented as a function of the dilution level, indicated by the integrated $CO_2$ area. Generally, a higher integrated $CO_2$ area suggests a more concentrated plume, leading to increased external OH and $O_3$ reactivity, which in turn reduces the concentration of OH radicals available in the Go:PAM for precursor oxidation (Emanuelsson et al., 2013; Watne et al., 2018). Some buses displayed primary emissions too dilute for detection (markers located to the left in Figure 2b) but still exhibited non-negligible $EF_{PM:aged}$ after oxidation. To further examine the effects of simulated atmospheric oxidation in the Go:PAM, an estimated minimum $OH_{exp}$ was calculated for each plume by incorporating the OH reactivities of CO and HC and the titration of $O_3$ with NO, following methodologies from Watne et al. (2018) and Zhou et al. (2021). For all plumes, $OH_{exp}$ varied between $1.1 \times 10^9$ to $4.6 \times 10^{11}$ molecules $cm^{-3}$ s. The $EF_{PM:aged}$ for some buses, for example, the DSL and HVO located to the right in Figure 2c, increased with increasing $OH_{exp}$. However, due to potential large differences in the chemical composition of emissions across different passages of the same bus, where some species are more prone to forming secondary particle mass even at lower $OH_{exp}$, the $OH_{exp}$ dependent $EF_{PM:aged}$ for other buses was less pronounced.'

**6.) Figure 3: The grouping of Delta-PM according OH exp in Fig. 3 should be explained. Where all buses measured at five different OHexp in the Go:PAM or do the groups represent different vehicle and fuel types? What is the explanation of the lower dPM at 1-5 OH days in the tunnel study by Tkacik et al., 2014?**

[R6] We apologize for any confusion caused.

[A6] The box plot in Figure 3 summarizes $\Delta$PM as a function of OH exposure ($OH_{exp}$), incorporating various bus types from Go:PAM measurements in this study. The composition of the buses includes 40% DSL, 12.2% CNG, 20% RME, 20.8% HVO, and 7% $HVO_{HEV}$. The results were grouped according to different ranges of $OH_{exp}$, covering from $1.1 \times 10^9$ to $4.6 \times 10^{11}$ molecules $cm^{-3}$ s. To clarify, we have revised line 339 as follows: 'Figure 3 illustrates $\Delta$PM as a function of $OH_{exp}$ for the bus fleet in this study, which includes 40% DSL, 12.2% CNG, 20% RME, 20.8% HVO, and 7% $HVO_{HEV}$. The results were grouped based on $OH_{exp}$, spanning a range from $1.1 \times 10^9$ to $4.6 \times 10^{11}$ molecules $cm^{-3}$ s.'

The secondary PM formation reported in Tkacik et al. (2014) involved a fleet predominantly composed of light-duty gasoline vehicles (at least 80%), which likely yields lower $\Delta$PM than observed in our study. We have included additional comparative data after line 362: 'The $\Delta$PM in this study was comparable to 855 mg kg-fuel$^{-1}$ for a mixed fleet consisting of 44.1% gasoline, 41.3% diesel, and 14.6% LPG vehicles measured at an urban roadside in Hong Kong (Liu et al., 2019b). It was slightly higher than the $\Delta$PM measured from a Euro VI dominated (more than 70%) heavy-duty truck fleet at an urban roadside in Gothenburg (Zhou et al., 2021), and from a fleet with over 80% light-duty gasoline vehicles in a Pittsburgh tunnel study (Tkacik et al., 2014).'

*7.) Table 2: a sulfur-containing compound with sum formula CH4SO3 is among the top 10 emission factors of fresh gaseous emissions. Is this methanesulfonic acid, CH4O3S? If yes, then elaborate more on it because it is usually known as a tracer of secondary biogenic organics in marine environments.*

[R6] [A6] Corrêa and Arbilla (2008) have noted that mercaptans are emitted in diesel and biodiesel exhaust, which are particularly reactive under high NOx concentrations. In such environments, NOx may catalyze the transformation of mercaptans into products including methanesulfonic acid. Additionally, diesel fuel and lubricants commonly contain sulfur-containing organic compounds that, upon combustion, can potentially form various sulfuric derivatives. The presence of catalytic converters in diesel engines might further facilitate chemical reactions leading to unexpected byproducts, as the high temperatures and catalytic surfaces within these systems could modify the transformation pathways of sulfur compounds. In our study, we identified $CH_4SO_3$, potentially methanesulfonic acid, in the emissions from DSL and RME buses. Given the complexity of these reactions and the conditions within diesel exhaust systems, a detailed pathway for the formation of $CH_4SO_3$ remains to be fully elucidated. To address this, we have incorporated the following text after line 424: '$CH_4SO_3$, potentially identified as methanesulfonic acid, was detected in the emissions from DSL and RME buses. Previous studies, such as those by Corrêa and Arbilla (2008), have shown that mercaptans, emitted from diesel and biodiesel exhausts, can transform under high NOx

conditions into products including methanesulfonic acid. The presence of sulfur-containing organic compounds in diesel fuel and lubricants, and their potential transformation upon combustion into various sulfuric derivatives, alongside the catalytic activity of engine converters, could also contribute to such findings. However, the detailed formation pathway of $CH_4SO_3$ in our study remains unknown."

***8.) Conclusion/ atmospheric implications (P17, line 389-392): It is obvious that the omission of the secondary formation of particulate matter in engine exhaust in current legislation is problematic for understanding the potential impacts of mobile sources. However, it is unclear how the secondary PM from mobile sources should be implemented in regulations, given that dPM measurements depend on a variety of environmental factors that cannot be controlled, such as OHexp and dilution. The authors should present recommendations how to standardize the comparison of dPM measured at different OHexp.***

[R8] Thank you for your insightful suggestions. For a more detailed response, please refer to [R0].

[A8] We have included the following discussion in our manuscript after line 512: 'Despite more than an order of magnitude difference in $EF_{PM:Fresh}$ among buses operated with various fuel types, we observed smaller variations in $EF_{PM:Aged}$, suggesting that secondary particle formation is likely influenced by substantial non-fuel-dependent precursor sources such as lubrication oils and/or fuel additives. Recognizing these sources is crucial for refining regulations on hydrocarbon emissions, which could notably enhance SOA control. The median ratios of aged to fresh particle mass emission factors, listed in ascending order, were for diesel (4.0), HVO (6.7), $HVO_{HEV}$ (10.5), RME (10.8), and CNG buses (84), highlighting the significant yet often overlooked contributions of aged/photochemically processed emissions to urban air quality. Furthermore, Zhao et al. (2017) revealed a strongly nonlinear relationship between SOA formation from vehicle exhaust and the ratio of non-methane organic gas to $NO_x$ (NMOG:$NO_x$). For instance, increasing the NMOG:$NO_x$ from 4 to 10 ppbC/ppbNO$_x$ increased the SOA yield from dilute gasoline vehicle exhaust by a factor of 8, underscoring the importance of integrated emission control policies for $NO_x$ and organic gases for better manage SOA formation. While implementing regulations for secondary particle formation presents significant challenges, these are crucial for a thorough understanding of their impact on regional air quality and health. Our approach to measuring the maximum SOA formation potential—peaking at a photochemical age of approximately 5 equivalent days of atmospheric OH exposure—provides a possible semi-quantitative reference for comparing SOA formation potential across different studies. We acknowledge the limitations of this approach for direct regulatory application and emphasize the need for more precise and comprehensive research to develop a methodologically robust framework that stakeholders can agree upon for systematically assessing the impacts of vehicle on air quality and informing regulatory strategies.'

*9.) Conclusion/ atmospheric implications (P17, line 396-400): Please elaborate more on the atmospheric implications of HNCO from city buses, e.g. chemical reactions in the atmosphere, relevance of urea-SCR exhaust systems in different buses and bus fleets, and expected concentration in street environments. For example, relate the results of this study on bus emissions of isocyanic acid to the study by Jathar et al. (2017) who investigate diesel engines as an atmospheric source of isocyanic acid in urban areas.*

[R9] [A9] Thank you for your suggestions. In our original manuscript, we provided a comparison of HNCO EF with findings from Jathar et al. (2017) and other studies on line 414. We noted, 'Isocyanic acid, likely an intermediate product of the thermal degradation of urea in SCR systems without sufficient hydrolysis (Bernhard et al., 2012), was detected in emissions from all bus types, with $^{Md}$EFs of 0.08-14.74 mg kg-fuel$^{-1}$. These values are slightly lower than those from a non-road diesel engine (31-56 mg kg-fuel$^{-1}$) reported by Jathar et al. (2017), but align well with SCR-equipped diesel vehicles tested by Suarez-Bertoa and Astorga (2016) (1.3-9.7 mg kg-fuel$^{-1}$) and a diesel engine with a diesel oxidation catalyst (DOC) (Wentzell et al., 2013) (0.21-3.96 mg kg-fuel$^{-1}$). Among all Euro V/EEV buses, HVO$_{HEV}$ buses showed the highest emissions of HNCO, potentially attributed to cold engine conditions since the combustion engine does not operate continuously.'

Furthermore, we have expanded our discussion on the atmospheric implications of HNCO emissions and incorporated the following revisions after line 544 in our manuscript: 'Furthermore, there is a risk that some abatement systems might generate unintended compounds, such as HNCO from the thermal degradation of urea in SCR systems without sufficient hydrolysis. Additionally, Jathar et al. (2017) observed substantial direct emissions of HNCO from diesel engines and estimated that ambient concentrations in Los Angeles could vary widely, ranging from 20 to 107 ppt depending on different parameterizations of diesel engine emissions. The persistence of HNCO in the atmosphere, particularly under dry conditions, poses significant health risks. It has been linked to severe outcomes including respiratory and cardiovascular disorders, atherosclerosis, cataracts, and rheumatoid arthritis (Leslie et al., 2019; Roberts et al., 2011).'

[revised manuscript text omitted]

---

## Author Comment (AC2)

**Responses to reviewers**

Reviewer comments are in in ***black italic type***. Author responses are labelled with [R] and authors actions with [A]. Line numbers in the responses correspond to the **revised manuscript with track-changes**. Modifications to the manuscript are in blue.

*Reviewer 2*

***This paper studies the primary and secondary emissions of buses with a variety of different modern fuels, using a combination of gaseous, PM and CIMS measurements. This is potentially important as people switch from traditional fossil fuels to more sustainable fuels and buses often lead the way on this, mandated by local authorities.***

***Generally, ACP's remit would normally be concerned with atmospheric processes and implications rather than emissions profiling, however given the chemistry associated with the secondary production, I would see this an in-scope. That said, the paper is very much focused on the results rather than the atmospheric implications. But the data that is presented seems to have been collected and treated in an appropriate manner. It should be noted that the use of OFRs is not established as a perfect simulation for atmospheric processes (some of the shortcomings are referred to in the paper), but still should be taken as an indication that secondary aerosols are worthy of concern. The manuscript is generally well written, although it does rely too much on acronyms and symbols, making the paper unclear in places. I recommend publication after minor corrections.***

[R0] We greatly value the reviewer's insightful feedback and constructive suggestions.

[A0] We have revised our manuscript to more clearly articulate the implications of secondary particle formation on urban air quality and the complexities of regulating such emissions across different fuel types and engine technologies. Our study highlights the potential significance of non-fuel-dependent sources like lubrication oils and fuel additives, which considerably influence secondary particle formation. Additionally, we have broadened our discussion to include a possible semi-quantitative reference for comparing SOA formation potential across diverse studies. This enhanced discussion is now detailed after line 512 in our manuscript: 'Despite more than an order of magnitude difference in $EF_{PM:Fresh}$ among buses operated with various fuel types, we observed smaller variations in $EF_{PM:Aged}$, suggesting that secondary particle formation is likely influenced by substantial non-fuel-dependent precursor sources such as lubrication oils and/or fuel additives. Recognizing these sources is crucial for refining regulations on hydrocarbon emissions, which could notably enhance SOA control. The median ratios of aged to fresh particle mass emission factors,

listed in ascending order, were for diesel (4.0), HVO (6.7), HVO$_{HEV}$ (10.5), RME (10.8), and CNG buses (84), highlighting the significant yet often overlooked contributions of aged/photochemically processed emissions to urban air quality. Furthermore, Zhao et al. (2017) revealed a strongly nonlinear relationship between SOA formation from vehicle exhaust and the ratio of non-methane organic gas to NO$_x$ (NMOG:NO$_x$). For instance, increasing the NMOG:NO$_x$ from 4 to 10 ppbC/ppbNO$_x$ increased the SOA yield from dilute gasoline vehicle exhaust by a factor of 8, underscoring the importance of integrated emission control policies for NO$_x$ and organic gases for better manage SOA formation. While implementing regulations for secondary particle formation presents significant challenges, these are crucial for a thorough understanding of their impact on regional air quality and health. Our approach to measuring the maximum SOA formation potential—peaking at a photochemical age of approximately 5 equivalent days of atmospheric OH exposure—provides a possible semi-quantitative reference for comparing SOA formation potential across different studies. We acknowledge the limitations of this approach for direct regulatory application and emphasize the need for more precise and comprehensive research to develop a methodologically robust framework that stakeholders can agree upon for systematically assessing the impacts of vehicle on air quality and informing regulatory strategies.'

**1. *The paper's title and overview would give the impression that this is a more comprehensive measurement set than it is, but it must be stressed that the acetate-CIMS method is very selective towards polar molecules such as organic acids. I would suggest modifying the title to something like "... primary and secondary emissions of particulate matter and polar molecules ..." or similar to better reflect the content of the paper. Also, the part dealing with the CIMS measurements (3.2) seems based around what the CIMS is capable of seeing, rather than what is important for the atmosphere. It would help the paper to remain in-scope if the discussion could be based around the importance of the subset of molecules observed in atmospheric chemistry, rather than merely observing what was seen by the CIMS.***

[R1] We value the insightful feedback from the reviewer and acknowledge the need to more clearly address the selective nature of the acetate-CIMS method towards polar molecules such as organic acids.

[A1] To better convey the scope and selective detection capabilities of our measurement set, we have revised the title to 'Online characterization of primary and secondary emissions of particulate matter and acidic molecules from a modern fleet of city buses.' This revision aims to clarify the focus on specific types of emissions analyzed in the study. Additionally, we have refined the abstract to highlight the selective ionization characteristics of the acetate reagent ion used in our CIMS setup. The updated sentences, now inserted after Line 25, read: "Online chemical characterization of gaseous and particulate emissions from

these buses was conducted using a chemical ionization mass spectrometry (CIMS) with acetate as the reagent ion, coupled with a filter inlet for gases and aerosols (FIGAERO). Acetate reagent ion chemistry selectively ionizes acidic compounds, including organic and inorganic acids, as well as nitrated and sulfated organics."

Further, in Section 3.2—Chemical characterization using CIMS—we focus on the acidic compounds identifiable by acetate-CIMS, which hold significant atmospheric relevance. To emphasize the importance of studying these acids, we have added the following text in the introduction after Line 138:'Primary emissions can also be oxidized to higher-volatility products through fragmentation reactions, potentially producing carboxylic acids (Friedman et al., 2017). Engine exhaust is a recognized primary source of organic and inorganic acids in urban environments (Kawamura et al., 1985; Kawamura and Kaplan, 1987; Kirchstetter et al., 1996; Wentzell et al., 2013; Friedman et al., 2017). Monocarboxylic acids are produced by both diesel and spark-ignited engines (Zervas et al., 2001a; Crisp et al., 2014; Zervas et al., 2001b; Kawamura et al., 1985). Recent studies have identified gaseous dicarboxylic acids in diesel exhaust (Arnold et al., 2012), compounds likely linked to the nucleation and growth of particles (Pirjola et al., 2015; Zhang et al., 2004). Additionally, inorganic acids such as nitric ($HNO_3$) and nitrous (HONO) acids, along with isocyanic acid (HNCO)—implicated in serious health issues like atherosclerosis, cataracts, and rheumatoid arthritis through carbamylation reactions—have been identified in both diesel and gasoline exhausts (Wang et al., 2007; Roberts et al., 2011; Wentzell et al., 2013; Brady et al., 2014; Link et al., 2016; Li et al., 2021). However, the secondary production of organic acid from engine exhaust remains poorly characterized; and it may significantly contribute to the overall organic acid budget and help explain discrepancies between models and measurements (Millet et al., 2015; Yuan et al., 2015; Paulot et al., 2011). Furthermore, the impacts of evolving fuel and engine technologies on emissions have not been comprehensively assessed. Recent advancements in analytical techniques now enable simultaneous, high-resolution online measurements of both gas and particle phase acidic species. This is facilitated by high-resolution time-of-flight chemical ionization mass spectrometry (HR-ToF-CIMS) using acetate as the reagent ion, coupled with a filter inlet for gases and aerosols (FIGAERO) (Le Breton et al., 2019; Friedman et al., 2017; Lopez-Hilfiker et al., 2014).'

*2. Presentation, wise, I found parts of the paper over-reliant on acronyms and symbols, particularly in the figure captions, where I found myself having to jump back and forward several times to understand what was being talked about. It would make the manuscript much clearer if the descriptions could be given verbally more.*

[R2] [A2] Thank you for highlighting this concern. We have revised the manuscript to enhance clarity by expanding the detailed textual descriptions and reducing reliance on acronyms and symbols in figure captions and text. This should improve readability and accessibility throughout the manuscript.

**3. *Section 3.3.2: It is worth noting that many of the gaseous emissions measured can also directly form secondary particulate matter in the atmosphere in the presence of water vapour and a base (e.g. ammonia), so these could also contribute to the overall secondary PM yield (in theory).***

[R3] [A3] We acknowledge the potential of gaseous emissions to directly contribute to the secondary particulate matter (PM) yield in the atmosphere. To address this important aspect, we have expanded our discussion in Section 3.2.2 to include the potential of carboxylic acids, measured in our study, to participate in secondary PM formation when interacting with water vapor and ammonia in the atmosphere.

We have added the following discussion after Line 456, 'It is important to note that many of these carboxylic acids can directly participate in secondary PM formation in the atmosphere in the presence of water vapor and a base such as ammonia (Chen et al., 2020; Huang et al., 2018; Hao et al., 2020). This process may significantly contribute to the overall secondary PM yield, reflecting a more complex interplay between gaseous emissions and particulate matter under atmospheric conditions.'

**4. *Line 401: The phrase " nitrogen-containing compounds were significantly reduced" is an unfortunate choice of words, because it is not clear whether "reduced" is in the magnitude or chemical sense. Please rephrase.***

[R4] We appreciate the reviewer's attention to detail and the potential for ambiguity in our phrasing.

[A4] To eliminate ambiguity, we have revised the phrase to specifically address the context of emission levels. The revised sentence in the manuscript now reads: "Notably, the emission levels of nitrogen-containing compounds were significantly lowered in Euro VI buses, equipped with advanced after-treatment systems that include EGR and DPF technologies in addition to SCR-only techniques." This modification appears in Line 401 and ensures clarity that we are discussing a decrease in emissions rather than a chemical reduction process.

**References**

[revised manuscript text omitted]